# Osteopontin is a therapeutic target that drives breast cancer recurrence

Yu Gu[1,2], Tarek Taifour[1,3], Tung Bui[1,2], Dongmei Zuo[1], Alain Pacis[1,4], Alexandre Poirier [1,3], Sherif Attalla[1], Anne-Marie Fortier [1], Virginie Sanguin-Gendreau[1], Tien-Chi Pan[5], Vasilios Papavasiliou[1], Nancy U. Lin [6], Melissa E. Hughes[6], Kalie Smith[6], Morag Park [1,2,7], Michel L. Tremblay [1,2,3,7], Lewis A. Chodosh[5], Rinath Jeselsohn [6] & William J. Muller [1,2,7] ✉

Recurrent breast cancers often develop resistance to standard-of-care therapies. Identifying targetable factors contributing to cancer recurrence remains the rate-limiting step in improving long-term outcomes. In this study, we identify tumor cell-derived osteopontin as an autocrine and paracrine driver of tumor recurrence. Osteopontin promotes tumor cell proliferation, recruits macrophages, and synergizes with IL-4 to further polarize them into a pro-tumorigenic state. Macrophage depletion and osteopontin inhibition decrease recurrent tumor growth. Furthermore, targeting osteopontin in primary tumor-bearing female mice prevents metastasis, permits T cell infiltration and activation, and improves anti-PD-1 immunotherapy response. Clinically, osteopontin expression is higher in recurrent metastatic tumors versus female patient-matched primary breast tumors. Osteopontin positively correlates with macrophage infiltration, increases with higher tumor grade, and its elevated pathway activity is associated with poor prognosis and long-term recurrence. Our findings suggest clinical implications and an alternative therapeutic strategy based on osteopontin's multiaxial role in breast cancer progression and recurrence.

Cancer relapse remains the major hurdle in the clinical management of breast cancer. Local and metastatic recurrent tumors often foster therapy resistance, leading to disease lethality. During the stepwise progression of breast cancer, a proportion of tumor cells can enter a state of dormancy characterized by temporary cell cycle arrest before they reactivate and resume proliferation[1,2]. Tumor cell-intrinsic adaptations such as genetic mutations must occur to disrupt the equilibrium maintained within a dormant tumor cell[3]. Additionally, tumor environmental changes also contribute to shifting the balance in favor of malignant progression and recurrence[3,4]. Therefore, dormancy exit is a forcefully dynamic and multi-stage process orchestrated by tumor cell-intrinsic and microenvironmental drivers within a recurrent tumor.

Genetically engineered mouse models (GEMMs) that recapitulate key features of human disease have provided valuable insights for our understanding of the complexity of tumor microenvironmental interactions in breast cancer progression and recurrence[1,5]. The versatility of GEMMs has allowed us to dissect the contributions of immune cell populations that promote relapse within a recurrent

[1]Rosalind and Morris Goodman Cancer Institute, McGill University, Montreal, QC, Canada. [2]Department of Biochemistry, Faculty of Medicine and Health Sciences, McGill University, Montreal, QC, Canada. [3]Division of Experimental Medicine, Faculty of Medicine and Health Sciences, McGill University, Montreal, QC, Canada. [4]Canadian Centre for Computational Genomics, McGill University Genome Center, Montreal, QC, Canada. [5]Department of Cancer Biology, Perelman School of Medicine, University of Pennsylvania, Philadelphia, PA, USA. [6]Department of Medical Oncology, Dana-Farber Cancer Institute, Boston, MA, USA. [7]Faculty of Medicine, McGill University, Montreal, QC, Canada. ✉e-mail: william.muller@mcgill.ca

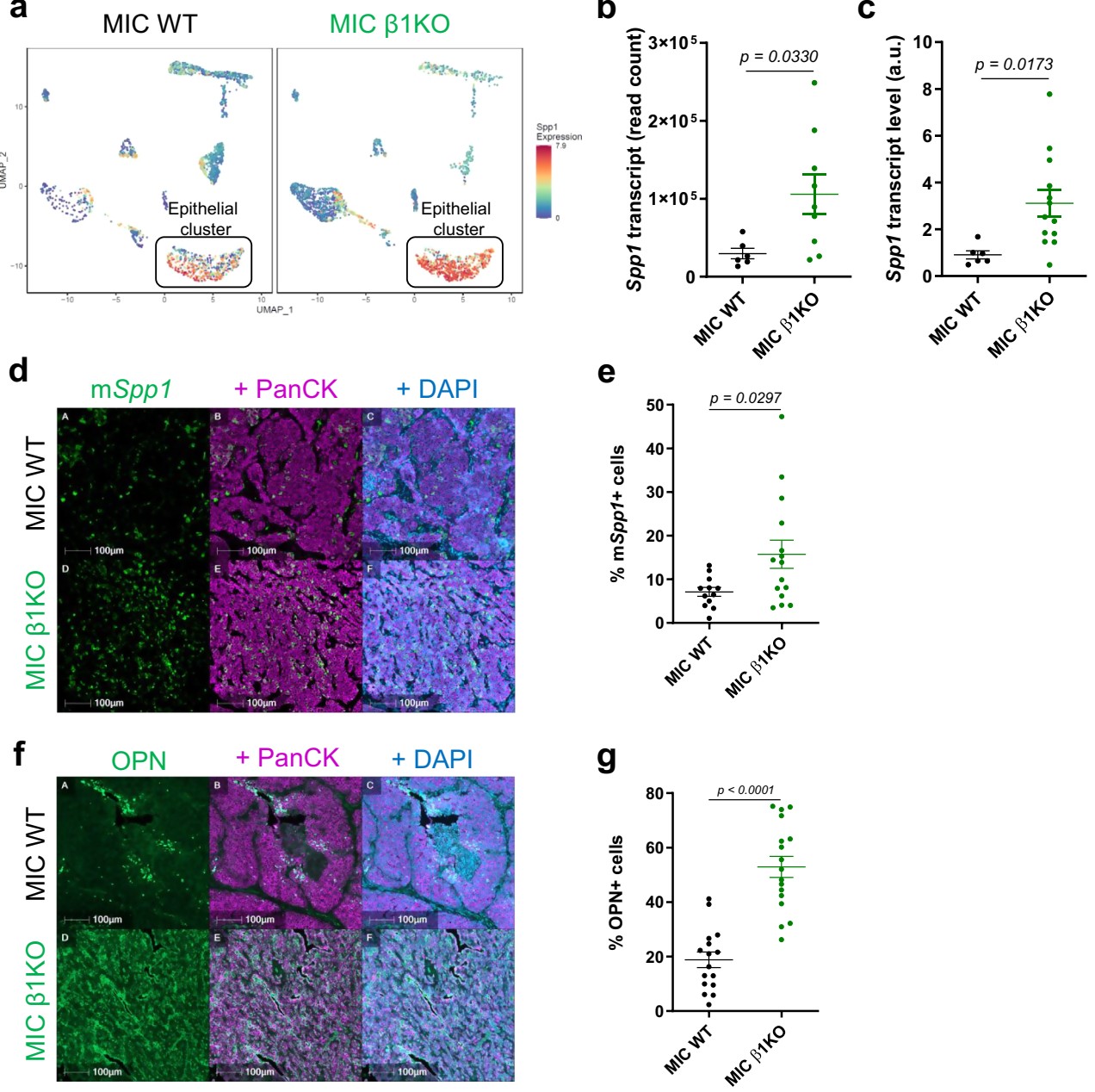

**Fig. 1 | β1 integrin-deficient recurrent tumors have elevated levels of osteopontin (OPN, *Spp1*). a** UMAP plots showing *Spp1* expression from single-cell RNA sequencing of early invasive carcinoma from MIC WT lesions (fast-growing, pooled lesions from *n* = 3 mice) or MIC β1KO lesions (dormant, pooled lesions from *n* = 6 mice), specifically in the epithelial tumor cell cluster. **b** Normalized read counts for *Spp1* from RNA-seq data from MIC WT (*n* = 6) or MIC β1KO (*n* = 9) recurrent tumors that exited dormancy from GSE186491. **c** RT-qPCR analysis for *Spp1* transcript, normalized to *Gapdh* from MIC WT (*n* = 6) and MIC β1KO (*n* = 13) recurrent tumors. **d** RNA Scope for mouse *Spp1* (m*Spp1*) and fluorescent immunohistochemistry

(IHC) for PanCK and DAPI on MIC WT (*n* = 12) and MIC β1KO (*n* = 15) recurrent tumors. **e** Quantification of m*Spp1*+ cells in MIC WT (*n* = 12) and MIC β1KO (*n* = 15) recurrent tumors. **f** Fluorescent IHC for OPN, PanCK, and DAPI on MIC WT (*n* = 16) and MIC β1KO (*n* = 17) recurrent tumors. **g** Quantification of OPN+ cells in MIC WT (*n* = 16) and MIC β1KO (*n* = 17) recurrent tumors. Scale bars are as indicated on each image. Mean ± SEM for data calculated using two-tailed Student's *t* test. Each data point is representative of one biological sample for (**b**), (**c**), (**e**), and (**g**). Source data are provided as a Source Data file.

tumor[1]. In addition to the tumor microenvironment (TME), GEMMs have provided critical insights into the role of cell adhesion receptor engagement in tumorigenesis. Integrins influence solid tumor progression by bridging intracellular signaling to environmental cues and vice versa[3,6,7]. β1 subunit-containing integrin dimers constitute the largest integrin subgroup as the exclusive collagen and laminin receptors and part of the RGD- and leukocyte-specific receptors[8]. Ablation of β1 integrin delays tumor onset and growth in vitro and in

vivo, with a notable impact on cell migration, invasion, and metastasis[3,6,7,9]. We have shown that mammary epithelium-targeted disruption of β1 integrin in a doxycycline-inducible Polyomavirus middle T antigen (PyV mT)-driven GEMM of breast cancer ("MIC" mouse) dramatically impaired mammary tumor development via senescence-mediated dormancy[3,5,10]. Yet, after a variable period of dormancy (6–28 weeks), 70% of MIC β1 integrin-deficient mice developed recurrent mammary tumors[3]. Therefore, they closely

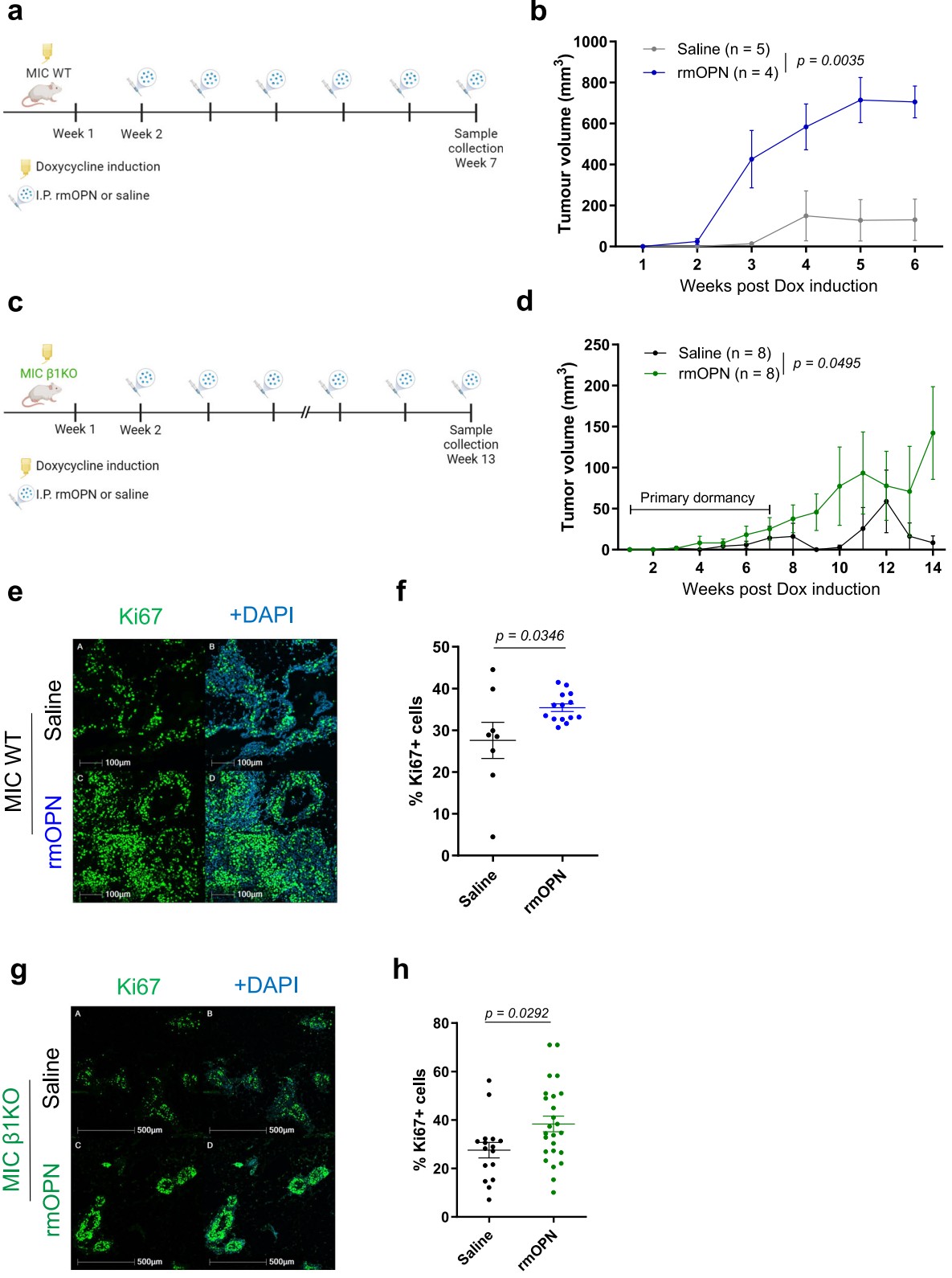

recapitulate many features of human breast cancer dormancy and recurrence. Our previous molecular analyses of the MIC β1 integrin-deficient recurrent tumors revealed both tumor cell-autonomous and TME alterations, including deposition of extracellular matrix (ECM) proteins[3]. ECM are an important group of acellular contributors to the tumor architecture and pro-tumorigenic signals for both epithelial tumor and stromal cells[11,12].

Studies across cancer types showed that the ECM osteopontin (OPN) promotes tumor cell growth and induces anti-apoptotic signaling through its activation of two classes of receptors, CD44 and integrin heterodimers[13–15]. OPN can further enhance tumor cell mobility, invasiveness, and epithelial-mesenchymal transition for dissemination and metastasis[16]. Most cells within solid tumors can respond to and secrete OPN although in varying concentrations and

**Fig. 2 | Exogenous osteopontin accelerates tumor growth in vivo. a** Schematic representation of experimental design for intraperitoneal injection of recombinant mouse OPN (rmOPN) supplementation to MIC WT mice induced on doxycycline. Created in BioRender. Muller, W. (2024) BioRender.com/k70t287. **b** Tumor volume measured from weekly palpations of MIC WT mice treated with saline or rmOPN. **c** Schematic representation of experimental design for intraperitoneal injection of rmOPN supplementation to MIC β1KO mice induced on doxycycline. Created in BioRender. Muller, W. (2024) BioRender.com/t18p112. **d** Tumor volume measured from weekly palpations of MIC β1KO mice treated with saline or rmOPN. Two-tailed Students' t-test was performed at endpoint and n denotes number of tumors per treatment arm for (**b**, **d**). **e** Fluorescent IHC for Ki67 and DAPI on tumors and mammary glands of MIC WT mice treated with saline ($n = 8$) or rmOPN ($n = 14$). **f** Quantification of Ki67+ cells in tumors and mammary glands of MIC WT mice treated with saline ($n = 8$) or rmOPN ($n = 14$). **g** Fluorescent IHC for Ki67, PanCK, and DAPI on tumors and mammary glands of MIC β1KO mice treated with saline ($n = 16$) or rmOPN ($n = 25$). **h** Quantification of Ki67+ cells in tumors and mammary glands of MIC β1KO mice treated with saline ($n = 16$) or rmOPN ($n = 25$). Scale bars are as indicated on each image. Mean ± SEM for data calculated using two-tailed Student's t test unless otherwise specified. Each data point is representative of one biological sample for (**f**, **h**). Source data are provided as a Source Data file.

isoforms[17,18]. OPN also harbors the potential to activate and to polarize stromal cells like macrophages and fibroblasts and can act as an immune checkpoint to suppress T cell activation, altogether setting a favorable tumor growth environment[17,19–21]. However, limited studies have specifically investigated OPN's role in the context of breast cancer recurrence although OPN contributes to numerous hallmarks for the survival of dormant tumor cells and their subsequent reactivation for recurrence[21–23]. Further in vivo and clinical validations are needed to decipher its mechanism and therapeutic potential.

Here, we report that OPN, a direct target of the signal transducer and activator of transcription 3 (Stat3), is elevated in MIC β1 integrin-deficient (MIC β1KO) recurrent tumors[24–26]. OPN promotes tumor cell proliferation in vitro and in vivo, recruits macrophages in a recurrent TME, and synergizes with tumor-derived IL-4 for them to acquire pro-tumorigenic characteristics. Macrophage depletion phenocopies OPN inhibition in reducing recurrent tumor burden. Furthermore, targeting OPN in primary mammary tumor-bearing mice effectively inhibits tumor growth and lung metastasis, the former due to T cell-mediated clearance. We further demonstrate that targeting OPN improves anti-PD-1 response. Meta-analyses on patient datasets show that high OPN levels and activated OPN pathway signaling correlate with worse relapse-free survival and increased invasiveness in the clinic. Between patient-matched primary and recurrent metastatic tumors, OPN levels are higher and positively correlate with macrophage infiltration in recurrent tumors. Taken together, our results argue that OPN is a key modulator for immune TME-dependent breast cancer recurrence and is a potential therapeutic target to prevent recurrence with additive effects to current immunotherapies.

## Results

### Osteopontin, a direct target of Stat3, is elevated in β1 integrin-deficient recurrent tumors

Following our in-depth investigation and functional validation of tumor cell-intrinsic adaptations during breast cancer recurrence in MIC β1 integrin-deficient (MIC β1KO) transgenic mice[3], we sought to further characterize the TME changes that mediate breast cancer recurrence. Single-cell RNA sequencing (sc-RNA seq) on early invasive carcinoma MIC wild-type (WT) and β1 integrin-deficient tumors (dormant stage) revealed that the epithelial tumor cells expressed elevated levels of osteopontin (OPN, *Spp1*) (Fig. 1a), a secreted cytokine and ECM protein that has both tumor cell-autonomous and stromal targets. Bulk RNA sequencing analysis on MIC WT and MIC β1 integrin-deficient recurrent tumors (end stage), *Spp1*-specific RT-qPCR, and RNA fluorescent in-situ hybridization confirmed that elevated levels of OPN transcripts were elevated and primarily derived from epithelial tumor cells in recurrent tumors (Fig. 1b–e). Consistently, fluorescent immunohistochemistry (IHC) showed a significant increase in OPN protein levels in MIC β1 integrin-deficient recurrent tumors (Fig. 1f, g). However, OPN protein levels in MIC β1 integrin-deficient mammary glands at 2 weeks post-induction and early invasive MIC β1 integrin-deficient dormant tumors remained comparable to their MIC WT counterparts, arguing that OPN protein level is only elevated upon tumor recurrence (Supplementary Fig. 2a–d).

Previous studies demonstrated that OPN is a direct target of the transcription factor Stat3, an important regulator in establishing an immune suppressive TME that is crucial during tumor recurrence[24–26]. Indeed, phosphorylated Stat3 (p-Stat3) was elevated in MIC β1 integrin-deficient recurrent tumors (Supplementary Fig. 1a–f). Further analyses on the sc-RNA seq data indicated that epithelial tumor cell cluster upregulates Stat3 most prominently amongst all other cell clusters during the early invasive stage before recurrence (Supplementary Fig. 1g). Consistent with these results, OPN levels in mammary glands from MIC Stat3-deficient mice at two weeks post-induction were significantly decreased compared to MIC WT mice (Supplementary Fig. 1h–j)[26]. Additionally, acute deletion of β1 integrin in MMTV-PyV mT *Itgb1* fl/fl cell lines increased Stat3 levels and OPN secretion (Supplementary Fig. 2e–h). Taken together, these data demonstrate that Stat3-dependent expression of OPN is notably increased in MIC β1 integrin-deficient recurrent tumors.

### Exogenous osteopontin accelerates tumor growth

To directly test whether OPN promotes tumorigenesis, we administered weekly intraperitoneal injections of recombinant mouse OPN into MIC WT mice after two weeks of induction (Fig. 2a). Weekly palpations showed that exogenous OPN was able to accelerate tumor growth (Fig. 2b). Fluorescent IHC analyses further demonstrated that exogenous OPN promotes tumor cell proliferation (Fig. 2e, f). Mice that received exogenous OPN had higher levels of OPN and macrophages in their mammary glands post-treatment (Supplementary Fig. 3a–e).

To validate that OPN is only exerting its pro-tumorigenic effects during the recurrent stage of tumor development, we repeated and extended the previous experiments in MIC β1 integrin-deficient mice after two weeks of induction (Fig. 2c)[1,3]. The results revealed that exogenous OPN was only able to accelerate tumor growth in MIC β1 integrin-deficient mice after a period of primary dormancy (Fig. 2d). Mammary glands and tumors post-treatment from MIC β1 integrin-deficient mice had increased cell proliferation (Fig. 2g, h). Given the intrinsically elevated levels of OPN in β1 integrin-deficient mammary glands, exogenous OPN did not further increase OPN deposition or macrophage levels post-treatment (Supplementary Fig. 3f–j). Taken together, our in vivo findings show that OPN induces cell proliferation and contributes to tumor growth in both MIC WT and MIC β1 integrin-deficient mice. However, exogenous OPN was not capable of overcoming primary dormancy.

### β1 integrin-deficient recurrent tumors have a favorable TME dominated by pro-tumorigenic macrophages

To further characterize the TME adaptations during cancer recurrence, we evaluated the immune landscape of MIC β1 integrin-deficient recurrent tumors. Consistent with elevated levels of phospho-Stat3 in MIC β1 integrin-deficient recurrent tumors, they harbored an increased number of F4/80+ and CD206+ pro-tumorigenic macrophages with arginase 1 expression (Fig. 3a–c, Supplementary Fig. 4a–c), although the total numbers of CD45+ immune cells, T cells, and neutrophils remained comparable to MIC WT tumors (Supplementary Fig. 4d–k). One of the cytokines that can skew macrophages

towards acquiring pro-tumorigenic characteristics is interleukin 4 (IL-4)[19]. Indeed, IL-4 was elevated in the MIC β1 integrin-deficient recurrent tumors (Fig. 3d, e). Moreover, IL-4+ cells' colocalization with pan-cytokeratin (PanCK) indicated that epithelial tumor cells are the main

source of IL-4 (Fig. 3f). Not only did we observe more pro-tumorigenic macrophages and IL-4 in MIC β1 integrin-deficient recurrent tumors but spatial analyses also revealed that they are located adjacent to proliferating epithelial tumor cells (Fig. 3g). There was a higher

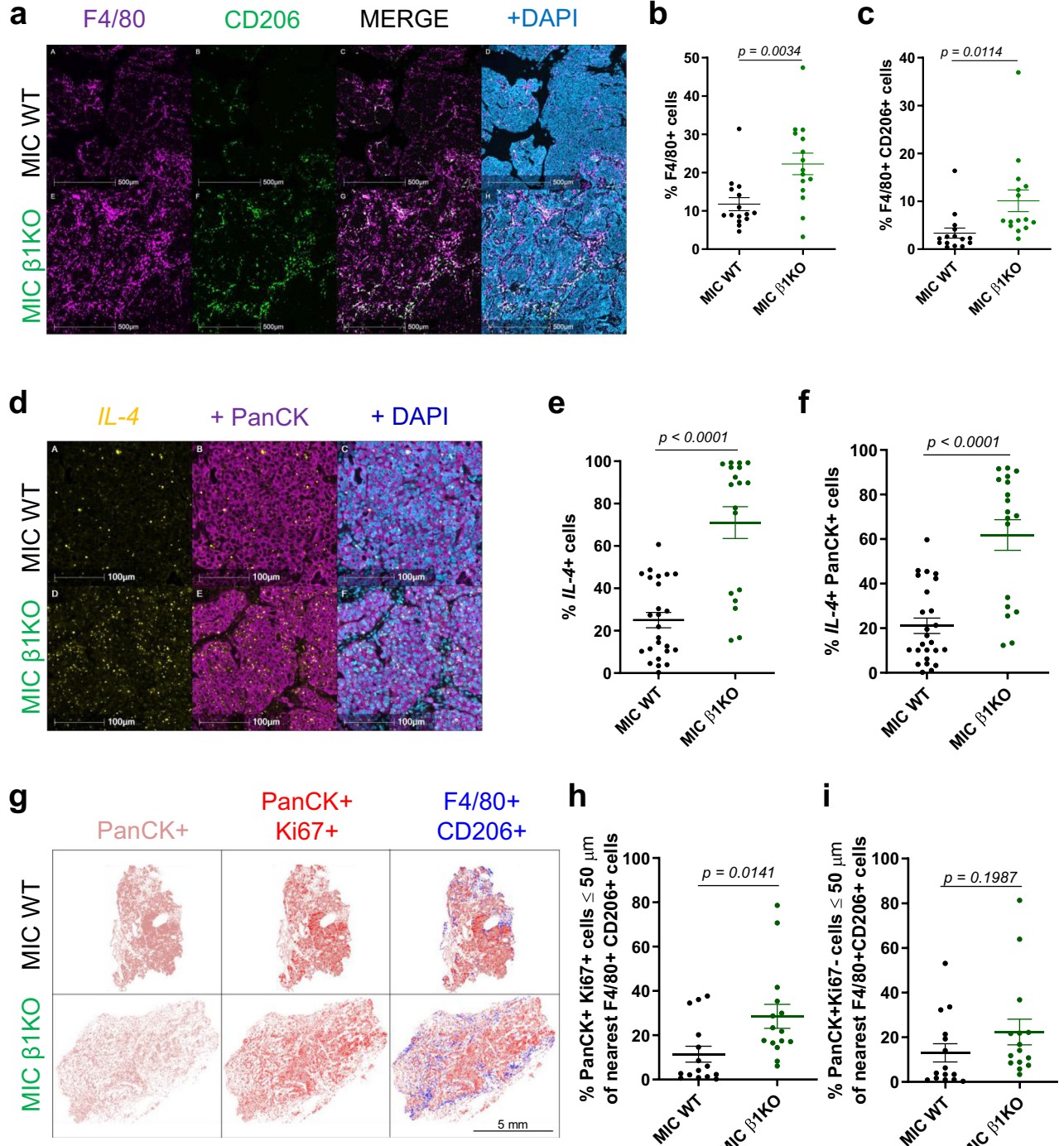

**Fig. 3 | β1 integrin-deficient recurrent tumors have elevated levels of pro-tumorigenic macrophages. a** Fluorescent IHC for F4/80, CD206, and DAPI on MIC WT (n = 15) and MIC β1KO (n = 15) recurrent tumors. **b, c** Quantification of total F4/80+ cells (total macrophages) and F4/80+ CD206+ cells (pro-tumorigenic macrophages) in MIC WT (n = 15) and MIC β1KO (n = 15) recurrent tumors. **d** RNA Scope for mouse *IL-4* and fluorescent IHC for PanCK and DAPI on MIC WT (n = 25) and MIC β1KO (n = 18) recurrent tumors. **e, f** Quantification of total *IL-4*+ cells and *IL-4*+ PanCK+ cells in MIC WT (n = 25) and MIC β1KO (n = 18) recurrent tumors. **g** Representative MIC WT (n = 15) and MIC β1KO (n = 15) recurrent tumors from (**a**)

for all PanCK+ epithelial cells, PanCK+ Ki67+ proliferating epithelial cells, and F4/80+ CD206+ macrophages. **h** Quantification of the percentage of PanCK+ Ki67+ cells within 50 μm of the nearest F4/80+ CD206+ macrophages in MIC WT (n = 15) and MIC β1KO (n = 15) recurrent tumors. **i** Quantification of the percentage of PanCK+ Ki67- cells within 50 μm of the nearest F4/80+ CD206+ macrophages in MIC WT (n = 15) and MIC β1KO (n = 15) recurrent tumors. Scale bars are as indicated on each image. Mean ± SEM for data calculated using two-tailed Student's *t* test. Each data point is representative of one biological sample for (**b**), (**c**), (**e**), (**f**), (**h**), and (**i**). Source data are provided as a Source Data file.

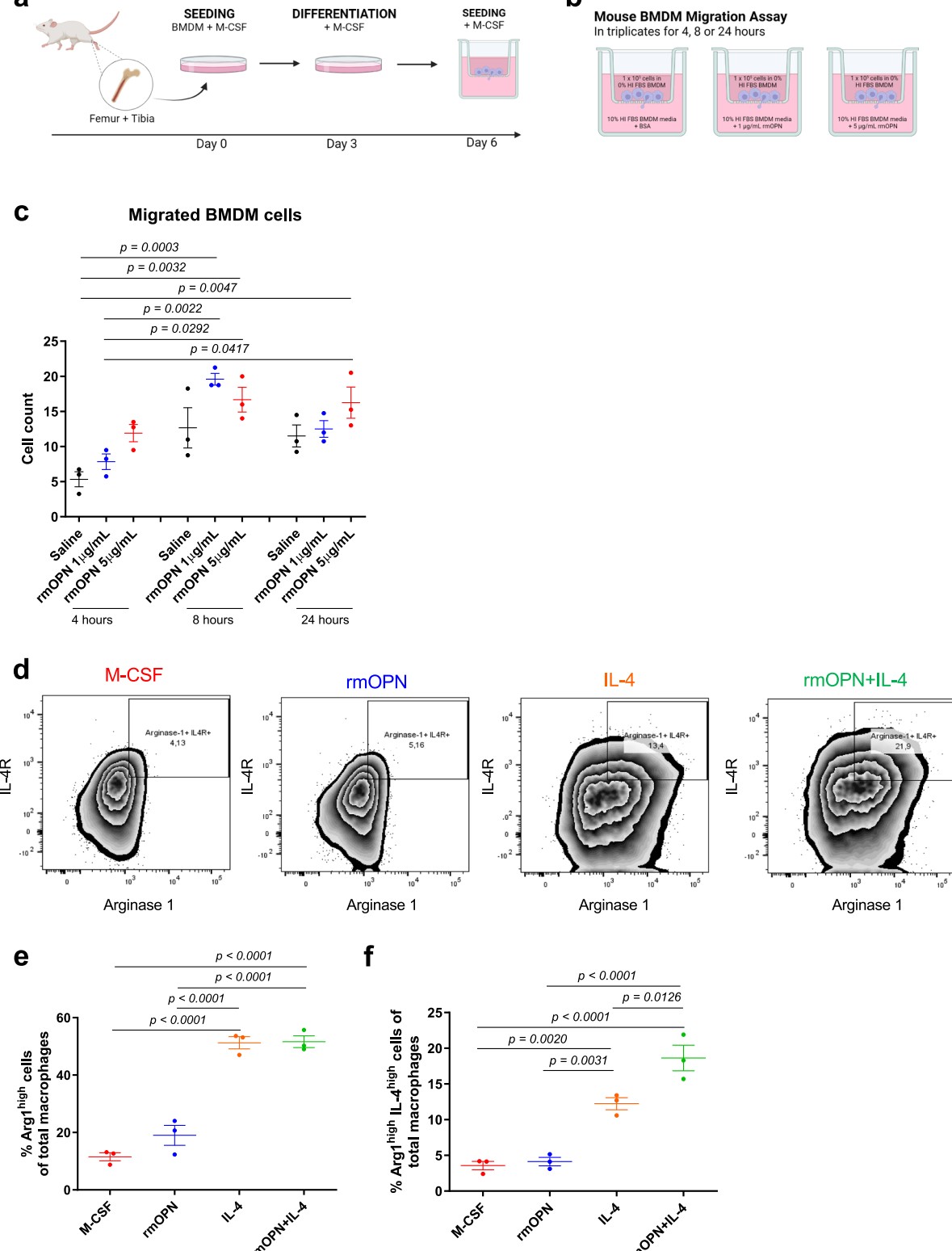

percentage of total Ki67+ PanCK+ proliferating tumor cells in closer proximity to pro-tumorigenic F4/80+ CD206+ macrophages in MIC β1 integrin-deficient recurrent tumors than in MIC WT tumors while the same analysis for non-proliferating tumor cells do not differ (Fig. 3h, i). These results argue that the pro-tumorigenic macrophages promote the proliferation of adjacent tumor cells in MIC β1 integrin-deficient recurrent tumors.

Consistent with the importance of OPN signaling axis, immunoblot analyses for OPN receptors (β3 integrin, αV integrin, and CD44) were significantly higher in MIC β1 integrin-deficient recurrent tumors (Supplementary Fig. 5a–d). To further delineate the cell population with elevated OPN receptor expression, we performed fluorescent IHC for OPN receptors. The results showed that the pro-tumorigenic macrophages in MIC β1 integrin-deficient recurrent tumors expressed

**Fig. 4 | Osteopontin induces macrophage migration and synergistically polarizes macrophages towards a pro-tumorigenic state with IL-4. a** Schematic representation of mouse bone marrow-derived macrophages (BMDMs). Created in BioRender. Muller, W. (2024) BioRender.com/w64j858. **b** Schematic representation of experimental design for migration assay of BMDM in vitro with various concentrations of recombinant mouse OPN (rmOPN). Created in BioRender. Muller, W. (2024) BioRender.com/h82t760. **c** Cell count of BMDM cells ($n = 3$ cell lines) in transwell migration assay towards saline, rmOPN at 1 and 5 µg/mL for 4, 8, and 24 h. Each data point is an average of four counts of four fields of

view. **d** FACS sorting for mouse BMDM cells ($n = 3$ cell lines) treated with saline, rmOPN, IL-4, and rmOPN and IL-4. Total F4/80+ macrophages from live cells sorted based on the expression of IL-4 receptor (IL-4R) and arginase 1. **e** Quantification of the percentage of arginase 1[high] cells out of total F4/80+ macrophages ($n = 3$ cell lines). **f** Quantification of the percentage of arginase 1[high] IL-4R[high] cells out of total F4/80+ macrophages ($n = 3$ cell lines). Mean ± SEM for data calculated using Ordinary One Way ANOVA with Tukey's post hoc test. Each data point is representative of one mouse BMDM cell line for (**e**, **f**). Source data are provided as a Source Data file.

elevated levels of OPN receptors in response to their high-OPN TME (Supplementary Fig. 5e–j). These data support the existence of paracrine signaling between OPN and macrophages in tumors supporting recurrence. Sc-RNA seq data further showed that macrophage clusters in MIC β1 integrin-deficient tumors expressed more ECM proteins in response to *Spp1*, namely *Col1a1, Col1a2, Col3a1, Dcn, Ecm1, Fnb1, Fn1,* and *Spp1* itself (Supplementary Fig. 6a, b), contributing the fibrotic recurrent TME as one of the well-documented functions of pro-tumorigenic macrophages[3,27,28]. Further sub-clustering analyses revealed that the majority of the macrophages are CD14/CD16 positive with higher CD206 (*Mrc1*) expression in the MIC β1 integrin-deficient tumors (Supplementary Fig. 6c–f). Collectively, our results demonstrate that increased OPN secretion by mammary epithelial tumor cells within recurrent tumors can act in a tumor cell-autonomous and paracrine manner to provide support for cancer recurrence.

### Osteopontin induces macrophage migration and synergizes with tumor cell-derived IL-4 for their polarization into a pro-tumorigenic state to promote recurrence

To investigate the mechanisms by which OPN is driving cancer recurrence through macrophages, we sought to assess if OPN is actively recruiting macrophages into the recurrent tumor in an isolated system given OPN's engagement with integrin receptors[13]. We collected and differentiated mouse bone marrow-derived macrophages (BMDMs) and seeded them in Transwell inserts with the presence of two different concentrations of recombinant mouse OPN (1 or 5 µg/mL) for 4 hours, 8 h or 24 h (Fig. 4a, b). BMDMs were then fixed and stained with crystal violet for counting. As expected, the presence of OPN did promote BMDM migration towards OPN-high media as early as 4 h after seeding (Fig. 4c).

Although our in vivo and in vitro observations demonstrate that OPN in recurrent TME recruits macrophages, it is unclear whether OPN can also function to polarize them to gain pro-tumorigenic characteristics. Currently, the field has not yet reached a consensus regarding OPN's effect on tumor-associated macrophage polarization[19,29,30]. To directly test whether OPN also contributes to macrophage polarization in addition to recruitment, we supplemented BMDMs with recombinant mouse OPN, IL-4, or recombinant mouse OPN and IL-4 together. Flow cytometry analysis showed that OPN alone was unable to polarize macrophages but OPN and IL-4 together further augmented the percentage of arginase 1+ and IL-4 receptor+ double-positive macrophages than those stimulated by IL-4 alone (Fig. 4d–f). Taken together, these data suggest that during breast cancer recurrence, tumor cell-derived OPN recruits macrophages into the tumor and, together with tumor cell-derived IL-4, further accentuates some of their pro-tumorigenic characteristics such as IL-4R expression to facilitate recurrence.

To directly validate the involvement of pro-tumorigenic macrophages in the outgrowth of MIC β1 integrin-deficient recurrent tumors, we used clodronate-liposomes to deplete macrophages in tumor-bearing MIC β1 integrin-deficient mice that already bypassed their primary dormant stage (Fig. 5a)[31,32]. The tumor volumes in treated mice remained relatively stable whereas tumors in PBS-liposome (control)-treated mice continued to grow until treatment endpoint (Fig. 5b). Histological analyses in the post-treatment mammary glands and tumors confirmed a notable decrease in total and pro-tumorigenic macrophages in clodronate-liposome-treated mice (Fig. 5c–e). This

was seen along with a decreased cell proliferation, confirming the pro-tumorigenic effects of these macrophages on epithelial tumor cell growth during recurrence (Fig. 5f–h). Of note, the levels of total and pro-tumorigenic macrophages in early invasive carcinoma MIC WT and β1 integrin-deficient tumors (dormant stage) did not differ, further supporting their role exclusively during recurrence (Fig. 5i–k).

### Targeting osteopontin decreases primary and recurrent tumor growth and lung metastasis, and improves response to immunotherapy

A significant clinical challenge in treating recurrent tumors lies within their resistance to the standard-of-care as they re-emerge from tumors that already withstood and survived an initial round of treatments. However, there are limited alternatives in the form of low-toxicity therapies[1,33,34]. The results from our study argue that neutralizing OPN could reduce tumor burden. To this end, we induced MIC WT mice for two weeks and administered six doses of mouse OPN-neutralizing antibody (anti-mOPN) or IgG control (Fig. 6a). Tumor growth significantly decreased in MIC WT mice treated with anti-mOPN, the result of which is equally recapitulated in syngeneic MIC β1 integrin-deficient recurrent tumors (Fig. 6b, Supplementary Fig. 7a, b). Furthermore, 44% (4 out of 9) of mice that received IgG had overt lung metastases whereas none of the anti-mOPN treated MIC WT mice (0 out of 6) had lung metastasis (Fig. 6c, d).

Further histological analyses in anti-OPN-treated MIC WT tumors revealed that they were heavily infiltrated with CD3 + CD4+ and CD3 + CD8 + T cells, the former with higher levels of PD-1 (Fig. 6e, f, Supplementary Fig. 7c–f). Characterization of the T cell population in anti-mOPN-treated tumors showed an enrichment in T cells expressing interferon-gamma (IFN-γ; *Ifng*) and granzyme B (Supplementary Fig. 8a–f). T cell exhaustion ligand PD-L1 and marker Tim3 did not significantly differ between anti-mOPN- and IgG-treated groups (Supplementary Fig. 8g–l). Altogether, our results support that targeting OPN relieves T cell infiltration and exclusion which often confers immunotherapy resistance[35]. To further investigate whether targeting OPN can improve T cell-based immunotherapy response, particularly anti-PD-1, we treated syngeneic PyV mT tumors with anti-mOPN, anti-PD-1, both, or IgG controls (Fig. 6g). Parallel to results from previous studies, anti-PD-1 monotherapy was insufficient in reducing tumor growth (Fig. 6h)[36–39]. Anti-mOPN alone reduced tumor growth along with an increase in CD3 + CD8+ granzyme B + T cell (Fig. 6h, Supplementary Fig. 9a–c). These effects were further amplified in the combination treatment arm (Fig. 6h, Supplementary Fig. 9a–c). Altogether, the results indicate that neutralizing OPN not only impacts tumor growth and metastasis but also enhances T cell infiltration and activation in anti-PD-1-resistant tumors.

### Osteopontin is elevated in recurrent metastatic tumors and correlated with decreased relapse-free survival in patients

To validate the clinical implications of our findings, we analyzed a collection of patient datasets and human breast cancer tissue microarrays for OPN expression. Analyses of the BCFGG Biobank GSE142767 dataset showed that tumors expressed more *Spp1* than in normal mammary tissue in patients (Fig. 7a). Consistent with these RNA analyses, fluorescent IHC on two human breast cancer tissue microarrays

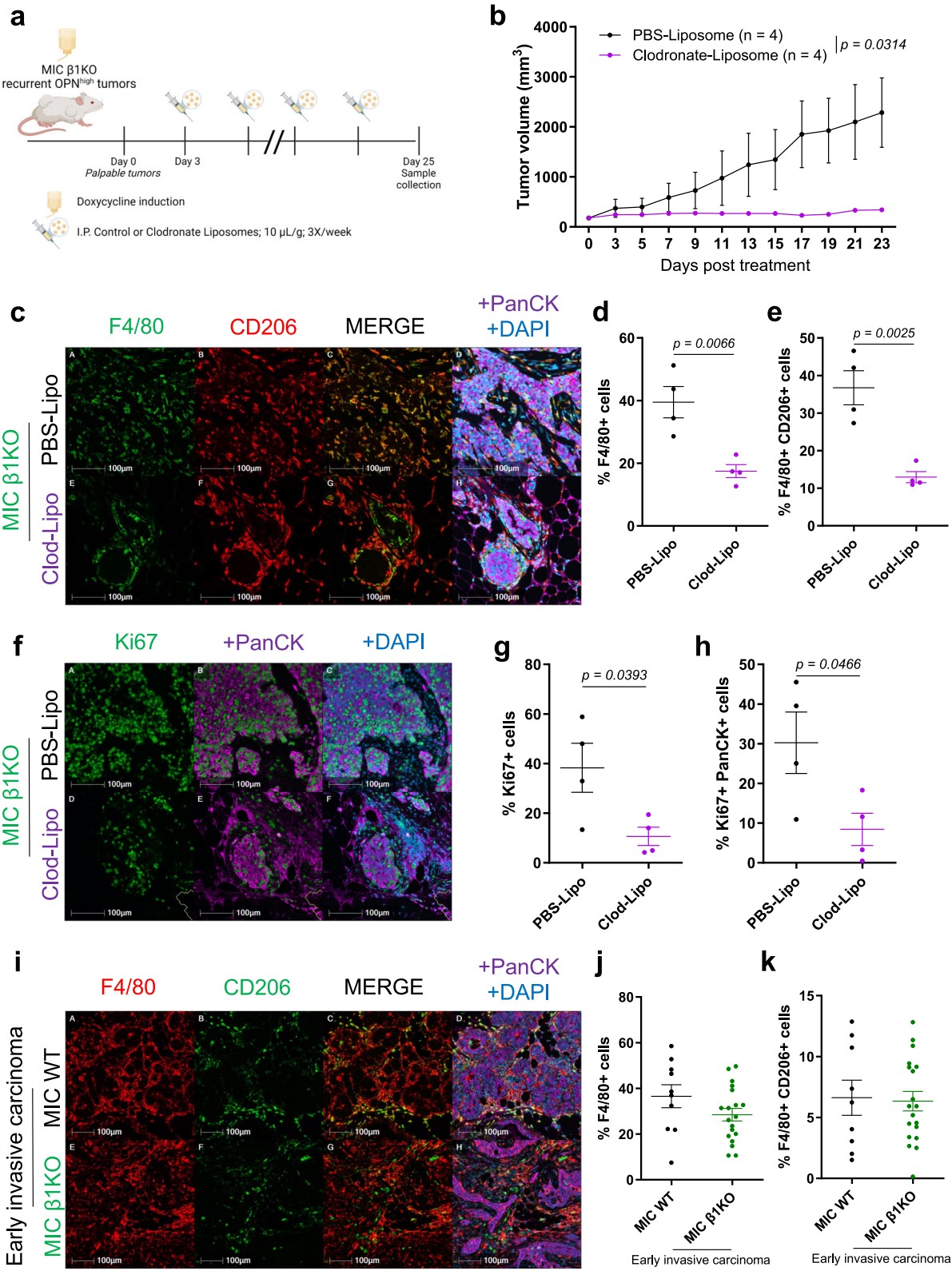

(BC081120f and BR1504b) for OPN revealed that 68% (13 out of 19) of adjacent normal tissue cores were OPN positive whereas 88% (200 out of 226) invasive carcinoma cores were OPN positive (Fig. 7b, c). Further analyses of the staining showed that invasive carcinoma tissues have significantly more OPN, both in the H score of OPN signal and in the percentage of cells expressing OPN with the majority of OPN+ cells co-localized with PanCK+ epithelial tumor cells (Fig. 7d, f). As the tumor

grade increased, we observed an increase in the H score of OPN signal and the percentage of OPN+ cells (total and epithelial tumor cell-specific) (Fig. 7g–i). These clinical data support that elevated OPN correlates with high-grade tumors with more likelihood of eventual recurrence and metastasis.

To further evaluate the role of OPN in breast cancer progression and recurrence, we performed fluorescent IHC for OPN on ten pairs of

**Fig. 5 | Macrophage depletion reduces MIC β1KO recurrent tumor growth.**
**a** Schematic representation of experimental design for intraperitoneal injection of
PBS- or clodronate-liposomes to MIC β1KO mice. Created in BioRender. Muller, W.
(2024) BioRender.com/r58v392. **b** Tumor volume measured from weekly palpa-
tions of MIC β1KO mice treated with PBS- or clodronate-liposomes. Two-tailed
Students' *t* test was performed at endpoint and n denotes number of tumors per
treatment arm. **c** Fluorescent IHC for F4/80, CD206, PanCK, and DAPI on tumors of
MIC β1KO mice treated with PBS- or clodronate-liposomes (*n* = 4 per group).
**d**, **e** Quantification of total F4/80+ cells and F4/80+ CD206+ cells in tumors of MIC
β1KO mice treated with PBS- or clodronate-liposomes (*n* = 4 per group).
**f** Fluorescent IHC for Ki67, PanCK, and DAPI on tumors of MIC β1KO mice treated

with PBS- or clodronate-liposomes (*n* = 4 per group). **g**, **h** Quantification of total
Ki67+ cells and Ki67+ PanCK+ cells in tumors of MIC β1KO mice treated with PBS- or
clodronate-liposomes (*n* = 4 per group). **i** Fluorescent IHC for F4/80, CD206,
PanCK, and DAPI on early invasive carcinoma from MIC WT lesions (fast-growing,
*n* = 10) or MIC β1KO lesions (dormant, *n* = 19). **j**, **k** Quantification of total F4/80+
cells and F4/80+ CD206+ cells in early invasive carcinoma from MIC WT lesions
(fast-growing, *n* = 10) or MIC β1KO lesions (dormant, *n* = 19). Scale bars are as
indicated on each image. Mean ± SEM for data calculated using two-tailed Student's
*t* test unless otherwise specified. Each data point is representative of one biological
sample for (**d**), (**e**), (**g**), (**h**), (**j**), and (**k**). Source data are provided as a Source
Data file.

patient-matched primary breast tumors and recurrent metastatic tumors
(Fig. 8a). Sites of metastasis include lymph node (*n* = 1), liver (*n* = 7), lung
(*n* = 1), and chest wall (*n* = 1) (Fig. 8a and Supplementary Fig. 10a).
Analyses revealed that OPN levels were significantly higher in patient-
matched recurrent metastatic tumors, both in signal intensity and per-
centage of OPN positivity, compared to primary breast tumors
(Fig. 8b–e). To assess whether elevated OPN resulted in a coordinated
increase in macrophage infiltration, we quantified the number of CD68+
macrophages in all the patient-matched samples (Supplementary
Fig. 10b). While the results indicated that recurrent metastatic tumors did
not have significantly more macrophages than their primary breast tumor
counterparts (Supplementary Fig. 10c), the percentage of OPN+ cells
positively correlated with CD68+ macrophage infiltration significantly
more in recurrent metastatic tumors than in primary breast tumors
(*p* = 0.0390 vs *p* = 0.2713, respectively) (Supplementary Fig. 10d, e).

We further analyzed Kaplan–Meier survival curves generated from
an invasive breast carcinoma dataset (GSE58644), which showed a
notable drop in the overall survival after 5 years for patients with high
*Spp1*-expressing tumors (Fig. 9a). We then consolidated an OPN pathway
and upregulated gene signatures from our findings (*Spp1*, *Timp3*, *Col11a1*,
*Mmp9*, *Mmp2*, *Fn1*, *Cd44*, and *Il-4*) to interrogate how their predicted
activity is associated with relapse-free survival in breast cancer patients.
We performed a meta-analysis on the hazard ratio of relapse-free survival
in breast cancer patients based on their estimated OPN levels and OPN
pathway activity (Fig. 9b)[40–45]. Indeed, there was a significant correlation
(*p* = 0.0051) between elevated OPN and OPN pathway activity, and higher
hazard ratio, reflective of a decrease in recurrence-free survival (Fig. 9b).
From gene signature analysis on invasive breast carcinoma stroma
(GSE9014), we observed a positive correlation between CD206+ pro-
tumorigenic macrophages and *Spp1* from the epithelial tumor compart-
ment (Fig. 9c). Lastly, TIMER 2.0 analysis of *Spp1* expression in various
solid tumors versus their adjacent normal tissue counterparts indicated
that in numerous cancers, including but not restricted to breast, cervical,
colon, esophageal, head and neck, liver, lung, skin, and stomach, *Spp1*
expression is significantly higher in tumor samples (Fig. 9d). Altogether,
these clinical validations are consistent with the findings in the in vivo and
in vitro models and further highlight OPN's multifaceted role in breast
cancer relapse and its therapeutic potential (Fig. 10a).

## Discussion

The ongoing efforts in developing alternative therapies to treat recurrent
tumors mainly stem from the compensatory mechanisms that recurrent
tumor cells engage in to confer resistance and eventually bypass the
initial anti-tumor efficacy of the standard-of-care regimes[46]. Therefore,
identifying pro-tumorigenic factors specifically in recurrent tumors
remains the rate-limiting step in improving survival for relapsing
patients. Here, we show that osteopontin (OPN) is a driver of breast
cancer recurrence through tumor cell-autonomous and immune mod-
ulations in the TME. Osteopontin is an effective therapeutic target to
reduce tumor burden and metastasis and a suitable additive to amelio-
rate immunotherapy response by relieving T cell exclusion.

Although OPN transcription begins to increase as early as the
primary tumor dormancy stage, its secreted protein levels are only

significantly elevated in recurrent tumors. Indeed, exogenous OPN
failed for β1 integrin-deficient tumors to bypass their primary dor-
mancy stage, yet increased cell proliferation to accelerate their sub-
sequent growth during recurrence. This demonstrates that OPN's pro-
tumorigenic effects should be most appreciated during recurrence.
Particularly, it is possible that the time until recurrence is necessary for
tumor cells to secrete and accumulate OPN within the TME to amplify
its pro-tumorigenic effect, especially in a paracrine fashion. Still, exo-
genous OPN is sufficient to promote cell proliferation and accelerate
tumor growth in a non-dormant model, and targeting OPN in the early
stages of tumor development remains effective in reducing tumor
growth and lung metastasis. These results underline its therapeutic
and anti-metastatic potential if given as an early treatment.

In addition to the acellular adaptations within recurrent TMEs, β1
integrin-deficient recurrent tumors revealed a noticeable increase in
OPN receptor+ macrophage infiltration recruited by OPN. This was in
concordance with an increase in IL-4 secretion by epithelial cells[19].
Furthermore, these macrophages acquire pro-tumorigenic character-
istics when stimulated with OPN and IL-4 to a CD206+ arginase 1+ pro-
tumorigenic state. They are spatially located near proliferating tumor
cells and further contribute to an ECM-rich TME. The expression of
OPN receptors on macrophages is vital to promote their migration and
thus recruitment into recurrent tumors, suggesting that in addition to
the accumulation of OPN in recurrent tumors, OPN receptor-
expressing stromal cells that can respond to the OPN-high TME are
equally active participants in promoting recurrence. This is supported
by our observations that OPN inhibition and macrophage depletion
phenocopied reduced tumor burden, functionally validating their
positive implications in recurrence. While the exact molecular path-
ways activated in macrophages by OPN are outside the scope of this
study, our in vitro data on BMDMs demonstrate that OPN stimulates
their migration. Several studies have elucidated the different pathways
activated by OPN, including Stat3- and integrin-activated
pathways[47–50]. It is noteworthy that these in vivo observations are
dependent on the oncogene driving mammary tumorigenesis and the
GEMM background. For example, the correlation between OPN and
macrophage infiltration was negligible in MMTV-c-myc/MMTV-v-Ha-
ras-driven mammary tumors[51]. Therefore, the unique set of immuno-
suppressive, chemotactic, and immune cell polarizing cytokines,
including but not limited to IL-4, secreted by PyV mT-driven mammary
tumor cells might not be reflected in other oncogene-driven tumors[5].

The early development of targeted therapies for breast cancers
initially deprioritized the adoption of immunotherapies as part of the
standard-of-care[52–54]. However, as residual tumor cells exploit alternative
hallmarks of cancer to confer resistance, the high relapse rate now
demands alternative treatment options[11,55–57]. To maximize the success of
immunotherapies, especially immune checkpoint inhibitors like anti-PD-
1, the immune TME must either be sensitive or re-sensitized with ade-
quate T cell infiltration[39,54,58,59]. Our study shows that the reduced tumor
burden from targeting OPN stems from cytotoxic T cell activity and
increased PD-1 levels. While the T cell suppression mediated by OPN has
previously been reported, limited studies validated its additive potency
to immunotherapies[21,60]. Here, we further demonstrate that in anti-PD-1-

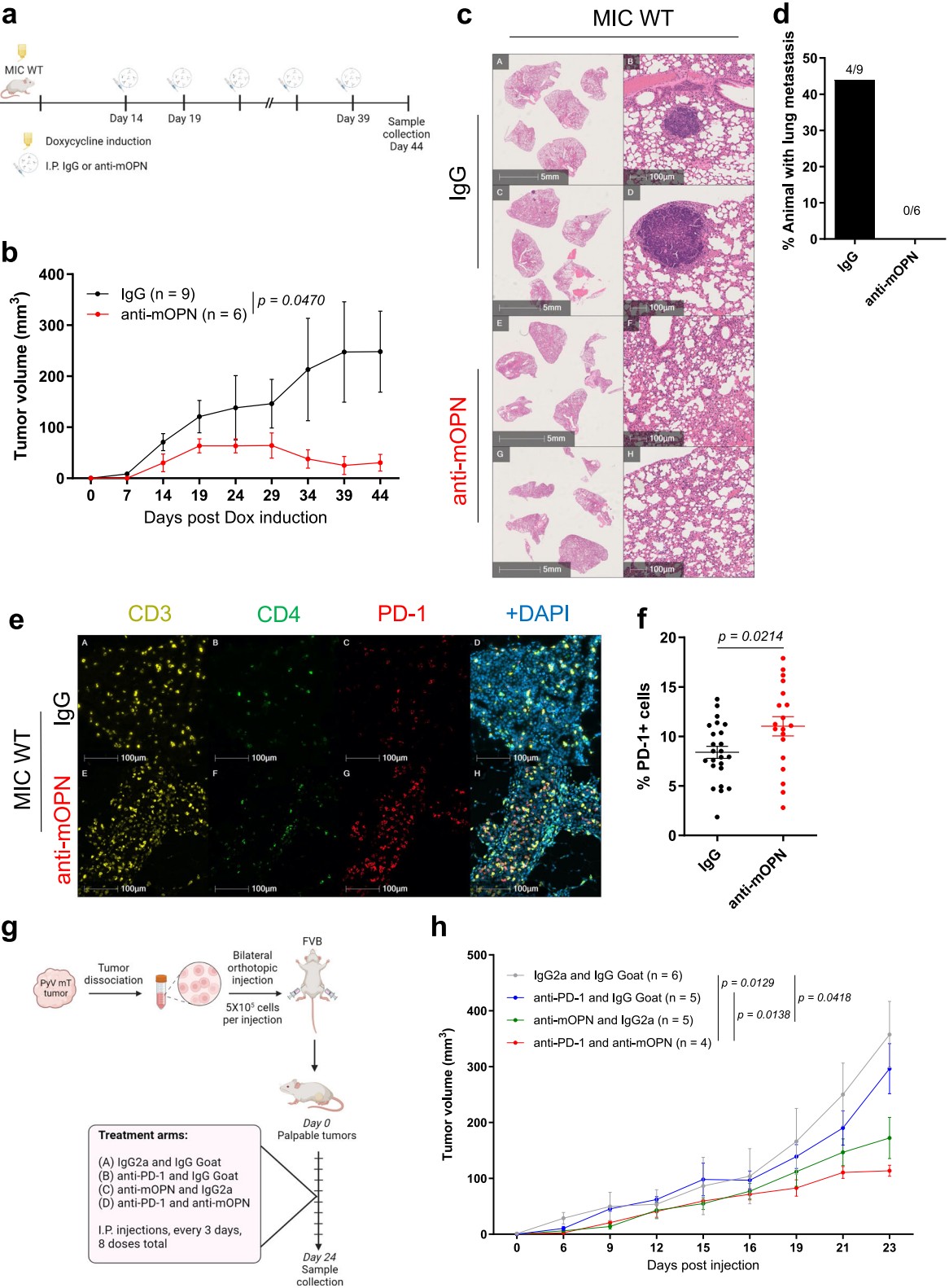

resistant syngeneic PyV mT tumors, anti-OPN and anti-PD-1 combination improves response compared to anti-PD-1 monotherapy[36–39]. This was attributed to higher T cell recruitment and activation which were previously shown to directly contribute to tumor cell death in this model[21,26,39]. It is noteworthy that the tumor growth was not significantly different between anti-OPN alone and the combinational treatment. Given the high clinical toxicity of anti-PD-1, these results suggest that anti-OPN alone could compensate for anti-PD-1 in anti-PD-1-resistant

OPN-positive tumors, though the clinical pharmacodynamics of anti-OPN remains to be investigated[61,62]. Our findings reinforce the importance of the stromal compartment in the TME and the potential benefits of leveraging ECM-targeted therapies in solid cancers.

Although integrin inhibitors have been explored as anti-cancer therapeutics with most efforts focused on targeting angiogenesis and metastasis, their clinical successes remain limited[46,63,64]. Notably, the large family of integrin heterodimers encourages compensatory

**Fig. 6 | Pharmacological inhibition of osteopontin decreases primary tumor growth and lung metastasis and improves anti-PD-1 response. a** Schematic representation of experimental design for intraperitoneal injection of control antibody IgG or mouse osteopontin neutralizing antibody (anti-mOPN) into MIC WT mice induced on doxycycline. Created in BioRender. Muller, W. (2024) BioRender.com/c88t325. **b** Tumor volume measured from weekly palpations of MIC WT mice treated with IgG or anti-mOPN. Two-tailed Students' t-test was performed at endpoint and n denotes number of tumors per treatment arm. **c** Representative H&E images of lungs in MIC WT mice treated with IgG ($n = 9$) or anti-mOPN ($n = 6$). **d** Percentage of animals with lung metastasis in MIC WT mice treated with IgG or anti-mOPN. **e** Fluorescent IHC for CD3, CD4, PD-1, and DAPI on tumors and mammary glands of MIC WT mice treated with IgG ($n = 24$) or anti-mOPN ($n = 19$). **f** Quantification of PD-1+ cells in tumors and mammary glands of MIC WT mice

treated with IgG ($n = 24$) or anti-mOPN ($n = 19$). **g** Schematic representation of experimental design for intraperitoneal injection of control antibodies IgG2a (control for anti-PD-1) and IgG Goat (control for anti-mOPN), anti-PD-1 and IgG Goat, anti-mOPN and IgG2a, or anti-PD-1 and anti-mOPN into FVB mice with mammary fat pad (MFP)-transplanted MMTV PyV mT tumors. Created in BioRender. Muller, W. (2024) BioRender.com/z45w893. **h** Tumor volume measured from weekly palpations of FVB mice with MFP-transplanted MMTV PyV mT tumors treated with IgG2a and IgG Goat, anti-PD-1 and IgG Goat, anti-mOPN and IgG2a, or anti-PD-1 and anti-mOPN. Two-tailed Students' t test was performed at endpoint and n denotes number of tumors per treatment arm. Scale bars are as indicated on each image. Mean ± SEM for data calculated using two-tailed Student's t test unless otherwise specified. Each data point is representative of one biological sample for (**f**). Source data are provided as a Source Data file.

mechanisms to sustain tumorigenesis, including upregulation of β3 integrin in the absence of β1 integrin[65]. Conceivably, β1 integrin-deficient epithelial cells undergo compensatory adaptation to upregulate OPN secretion to promote its engagement with other integrin receptors in an autocrine fashion[6,7]. However, our clinical data demonstrating OPN's upregulation in metastatic recurrent tumors, association with macrophage infiltration, and correlation with decreased relapse-free survival argue that OPN's involvement in recurrence is neither restricted to integrin modulation nor breast cancer alone. It is noteworthy that the increase in OPN-expressing epithelial tumor cells in metastatic tumors does not rule out the possibility that OPN is systemically elevated in these patients[13,14]. Several studies demonstrated that OPN promotes metastatic recurrence by establishing a permissive metastatic niche through other stromal cells prior to clinically detectable metastases[23,66,67]. Elucidating alternative TME targets such as OPN with global effects capable of re-structuring the immune and fibrotic landscapes could reveal alternative therapeutic strategies to address the clinical challenges posed by aggressive recurrent tumors and to improve long-term patient outcomes.

## Methods

### Ethics statement
All mice from animal experiments in this study were housed and handled at the Comparative Medicine and Animal Resource Centre at McGill University, approved by and in compliance with the Animal Ethics Committee, Facility Animal Care Committee, and Canadian Council on Animal Care (Animal Use Protocol #MCGL5518). All mice were housed at a maximum of five animals per cage with fluid and food ad libitum, on a 12 h dark/light cycle, at ambient temperature, and relative humidity of 45% to 65%. All mice were euthanized prior to or at the approved tumor volume endpoint of 2.5 cm³ for a single tumor mass or a total of 5 cm³ for multifocal tumors. Only female mice were used experimentally as this study pertains to female breast cancer.

### Animal model
MMTV-reverse tetracycline transactivator (rtTA) transgenic mice were generated in the laboratory of Dr. Lewis Chodosh as previously described[5,10]. Generation of MIC and MMTV-PyV mT transgenic mice was described previously, including *Itgb1* (β1 integrin) floxed alleles and Stat3 floxed alleles[3,26]. All mice were maintained on the pure FVB background. Genomic DNA was extracted from tails of all mice using crude salt extraction and subsequently used for genotype confirmation using PCR described previously[3]. Experimental and control animals were given drinking water with doxycycline (2 mg/mL) at 9 to 12 weeks of age (induction), weighed, and monitored weekly by physical palpations for tumor formation.

### Recombinant mouse osteopontin (rmOPN) treatment
MIC WT and MIC β1KO mice were induced for two weeks and given 5.0 µg of rmOPN (R&D Systems 441-OP) or saline weekly through intraperitoneal injections while on doxycycline. MIC WT mice were

given 6 doses while MIC β1KO mice were given 12 doses due to delayed tumor growth.

### Mouse osteopontin neutralizing antibody treatment
MIC WT mice were induced for two weeks and given 20 µg of mouse OPN neutralizing antibody (R&D Systems AF808) or 20 µg of Normal Goat IgG Control (R&D Systems AB-108-C) every five days through intraperitoneal injections while on doxycycline. MIC WT mice were given 6 doses. Doxycycline-induced FVB mice that received mammary fat pad (MFP) transplants of MIC β1KO recurrent tumors were given the same treatment, every three days, once tumors reached 50 mm³. Recurrent tumor transplant mice were given 10 doses.

### Mouse osteopontin neutralizing antibody and anti-PD-1 treatment
$5 \times 10^5$ MMTV-PyV mT cells were transplanted into FVB mice via mammary fat pad (MFP) injections. Once tumors reached 10 mm³, mice were given a combination of 20 µg of Normal Goat IgG (see above, control for anti-mOPN), 100 µg of Rat IgG2a (Bio X cell #BE0089, control for anti-PD-1), 20 µg of anti-mOPN (see above), and/or 100 µg of anti-PD-1 (RMP1-14, Bio X cell #BE0146) every three days through intraperitoneal injections. All mice were given 8 doses.

### Macrophage depletion treatment
MIC β1KO mice were induced until palpable "recurrent" tumors reached 50 mm³ after their primary dormancy from weekly palpations. Tumor-bearing mice were given 10 µL per gram of either PBS- or clodronate-liposomes (LIPOSOMA, Batch P03M0124 and C6M0224, respectively) three times per week through intraperitoneal injections. All mice were given 10 doses.

### Mouse tissue collection
Mammary gland, mammary tumor, and lungs were collected at various time points throughout this study. Mice were euthanized at end stage when an individual tumor or the total tumor mass reaches the end-point burden defined by McGill Animal Ethics Guidelines, or at various experimental or treatment endpoints. All solid organ tissues were fixed for 36 h in 10% (vol/vol) formalin (Leica), embedded in paraffin and sectioned at 4 µm for histological staining or were flash-frozen in liquid nitrogen and kept at −80 °C. H&E and immunohistochemistry-stained slides were scanned using Scanscope XT Digital Slide Scanner (Aperio Technologies) and analyzed using HALO 2.0 software (Indica Lab).

### Human samples
TMAs were obtained from TissueArray.Com (formerly US Biomax Inc) (BR1504b trial and BC081120f trial). Patient-matched primary breast tumor samples and recurrent metastatic tumor samples were obtained with IRB approval (Dana Farber/Harvard Cancer Center protocols 09-204 and 05–246) and with written patient consent. Samples are formalin-fixed paraffin-embedded tissues of ER-positive/HER2-

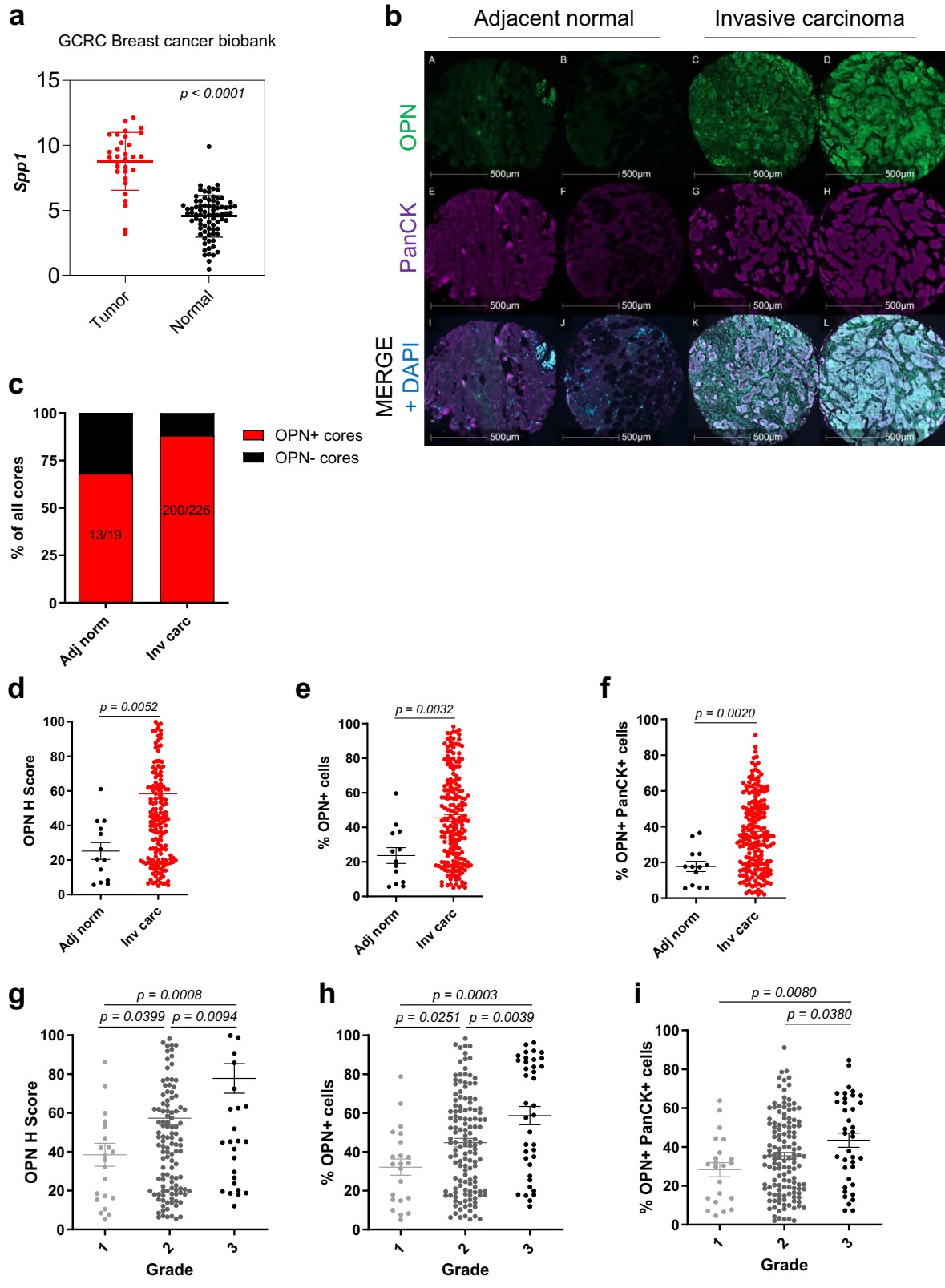

negative primary treatment-naïve breast cancers and matched metastatic tissue samples.

**Mouse mammary tumor single-cell and bulk RNA sequencing analysis**

Sample preparation and data analyses for sc-RNA seq and bulk RNA sequencing are previously published[3]. Datasets of MIC WT and β1KO

tumors are available on NCBI GEO GSE186118. Data analyses are performed by Alain Pacis from the Bioinformatics Core at McGill University as previously described[3].

**RNA extraction and RT-qPCR analysis**

Flash-frozen pieces of tumors were crushed in liquid nitrogen. Total RNA was isolated using FavorPrep™ Tissue Total RNA Mini Kit (Cat

**Fig. 7 | Human invasive carcinomas have elevated levels of osteopontin versus adjacent normal mammary tissue. a** *Spp1* expression in human breast tumor tissue (*n* = 31) and in normal human breast tissue (*n* = 80) from BCFGG Biobank GSE142767. **b** Fluorescent IHC for OPN, PanCK, and DAPI on primary human breast cancer tissue microarrays BC081120f and BR1504b (TissueArray.Com, formerly US Biomax). Representative images of samples are shown for adjacent normal cores and invasive ductal carcinoma cores. **c** Percentage of primary human breast cancer tissue microarray cores (BC081120f and BR1504b) with or without osteopontin expression in adjacent normal (Adj norm) and invasive ductal carcinoma (Inv carc) tissues. **d**–**f** Quantification of OPN H Score, total OPN+ cells, and OPN+ PanCK+ cells in adjacent normal (*n* = 13) and invasive ductal carcinoma (*n* = 200) cores in primary human breast cancer tissue samples. **g**–**i** Quantification of OPN H Score, total OPN+ cells, and OPN+ PanCK+ cells per grade in primary human breast cancer tissue samples (Grade 1: *n* = 22; Grade 2: *n* = 133; Grade 3: *n* = 38). Scale bars are as indicated on each image. Mean ± SEM for data calculated using two-tailed Student's *t* test. Each data point is representative of one biological sample for (**a**) and (**d**–**i**). Source data are provided as a Source Data file.

Number FATRK 001) according to manufacturer's protocol. RNA quantity was determined using NanoDrop Spectrophotometer ND-1000 (NanoDrop Technologies, Inc.). cDNA was synthesized by reverse transcription using the TranScript all-in-one first strand cDNA synthesis kit (Transgen Biotech). Real-time qPCR was performed using LightCycler 480 SYBR Green I Master Reagents (Roche). Data were normalized to *Gapdh* to generate the relative transcript levels using the expression $2^{(\text{crossing point value of } Gapdh - \text{crossing point value of gene of interest})}$. Each reaction was run in triplicate. The following primers were used for RT-qPCR analysis: *Spp1* - left primer: CAGCCTGCACCCAGATCCTA, right primer: GCGCAAGGAGATTCTGCTTCT; *Gapdh* - left primer: CTGCACCACCAACTGCTTAG, right primer: GTCTTCTGGGTGGCAGTGAT.

## Fluorescent immunohistochemistry (IHC), imaging, and quantitative analysis

Sample collection and preparation for paraffin-embedded mouse mammary gland and mammary tumor tissues were described above. Fluorescent IHC was performed as previously described[3]. Stained slides were scanned using the Axio Scan Z1 digital slide scanner (Carl Zeiss) and analyzed using HALO 2.0 software (Indica Lab). The same staining protocol was used for human samples. Unless otherwise specified, we have chosen to consistently use % Marker+ cells reflective of the percentage of total analyzed cells.

The following antibodies were used for fluorescent IHC on mouse tissue: PanCK (Ventana, 760–2595, 1:10), OPN (Santa Cruz, sc-21472, 1:1200), p-Stat3 (Cell Signaling Technology (CST), 9145, 1:100), F4/80 (CST, 70076, 1:200), CD206 (CST, 24595, 1:400), CD45 (CST, 70257, 1:200), CD3 (Abcam, Ab16669, 1:200), CD4 (CST, 25229, 1:50), CD8 (CST, 98941, 1:200), CD44 (CST, 37259, 1:200), β3 integrin (CST, 13166, 1:200), Ki67 (CST, 12202, 1:200), PD-1 (CST, 84651, 1:200), arginase 1 (CST, 93668, 1:100), α5 integrin (Santa Cruz, sc-376199, 1:100), granzyme B (CST, 44153, 1:200), PD-L1 (CST, 64988, 1:100), and Tim3 (CST, 83882, 1:200). The following antibodies were used for fluorescent IHC on human tissue: PanCK (Ventana, 760–2595, 1:10), OPN (Abcam, Ab63856, 1:100), and CD68 (Ventana, 790–2931, 1:5).

## RNA scope in situ hybridization

RNA Scope In-situ hybridization was performed on paraffin-embedded mammary tumor sections using RNAscope® 2.5 HD Assay-RED kit (ACD, #322360) according to the manufacturer's protocol. The following probes were used: Mouse-Mm-Spp1 (Cat Number 435191), Mouse-IL-4 (Cat Number 312741), and Mouse-Mm-Ifng (Cat Number 311391). For Fig. 1d, Fig. 3d, and Supplementary Fig. 8a, this protocol was followed with fluorescent IHC.

## Enzyme-linked immunosorbent assay (ELISA)

MMTV-PyV mT *Itgb1* fl/fl cell lines were infected with AdLacZ or AdCre in DMEM with EGF (5 ng/mL), bovine pituitary extract (35 µg/mL), insulin (5 µg/mL), hydrocortisone (1 µg/mL), penicillin (100 units/mL), streptomycin (100 µg/mL), gentamicin (50 µg/mL) supplemented with 5% vol/vol fetal bovine serum (FBS). Cell line generation and adenovirus infection were as described previously[3]. The supernatant was collected 7 days post-infection and diluted 1:20. Osteopontin (OPN/SPP1) Mouse ELISA Kit (Invitrogen EMSPP1) was used according to manufacturer's protocol.

## In vitro proliferation assay

MMTV-PyV mT cells were seeded in 96-well optical-bottomed plates (Nunc, 167008) at 8000 cells/well supplemented with recombinant mouse osteopontin at 25.0 µg/mL or bovine serum albumin (BSA) in triplicates. IncuCyte S3 system (ESSEN BioSciences) was used for live cell imaging at 10X for 2 images/well, every 4 h for 60 h. The confluence percentage was calculated using the IncuCyte S3 analysis software.

## Immunoblot sample preparation and analysis

Flash-frozen tumor pieces were crushed in liquid nitrogen and incubated in 300–500 µL of complete lysis buffer depending on pellet size (10 mM Tris-Cl pH 8.0, 1 mM EDTA, 0.5 mM EDTA, 1% Triton X-100, 0.1% sodium deoxycholate, 0.1% sodium dodecyl sulfate, 140 mM sodium chloride, 2 mM sodium pyrophosphate, 5 mM sodium fluoride, 10 mM β-glycerophosphate) with protease inhibitors (AEBSF 50 µg/mL, aprotinin 10 µg/mL, Leupeptin 10 µg/mL, $Na_3VO_4$ 100 µg/mL) for 1 h rotating at 4 °C, centrifuged at maximum speed for 15 min, and supernatant was collected. The protein concentration in supernatant was determined from OD reading by diluting in Protein Assay Dye (Bio-Rad) and calculated in reference to a BSA standard curve. Loading samples were prepared by mixing supernatant to a final protein concentration of 4 µg/µL, 6X protein loading buffer (375 mM Tris-HCl, 10% SDS, 35% Glycerol, 0.012% bromophenol blue, 9.3% DTT, 5% β-mercaptoethanol), and complete lysis buffer. All samples were denatured at 95 °C for 10 min and stored at −80 °C. Equal quantity of protein per sample was loaded on acrylamide gel for running at 120 V then transferred onto Immobilon®-FL PVDF transfer membrane for 90 min at 25 V at 4 °C. Membranes were blocked using Li-Cor Odyssey® Blocking Buffer (TBS) for 1 h at room temperature, incubated in primary antibodies overnight at 4 °C, washed in TBS with 1% Triton X-100, incubated in secondary antibodies (1:10,000) for 1 h at room temperature, washed, and imaged using Li-Cor Odyssey Scanner. Band intensity quantification was done using Image Studio Lite software (Li-Cor).

The following antibodies were used for immunoblots: Stat3 (CST, 9139, 1:1000), p-Stat3 (CST, 9145, 1:1000), β-actin (Sigma, A5441, 1:2000), tubulin (CST, 2148, 1:1000), β3 integrin (Abcam, Ab119992, 1:1000), αV integrin (CST, 60896, 1:1000) and CD44 (CST, 37259, 1:1000).

## Isolation and culture of mouse bone marrow-derived macrophages (BMDM)

Protocol adapted from ref. 68. Virgin female FVB mice were euthanized, and femurs and tibias were collected and kept on ice in 2% heat-inactivated FBS (HI FBS, FBS in 56 °C water bath for 30 min) in PBS. In the tissue culture hood, the epiphyses of femurs were twisted off and the bone marrow was flushed out using a 23 G needle and 3 mL syringe with 10% HI FBS BMDM media into a 6 well plate (BMDM media: DMEM with 1X glutamax, 1X sodium pyruvate, 1X β-mercaptoethanol, penicillin (100 U/mL), and streptomycin (100 mg/mL)). Likewise, both ends of tibias were cut off ad the bone marrow flushed out into the same well for the same mouse. Bone marrows were broken by pipetting up and down in 10% HI FBS BMDM media, passed through a 40 µm strainer and centrifuged for 5 min at 4 °C. All samples were treated with Ammonium-Chloride-Potassium (ACK) lysis buffer (150 mM NH$_4$Cl,

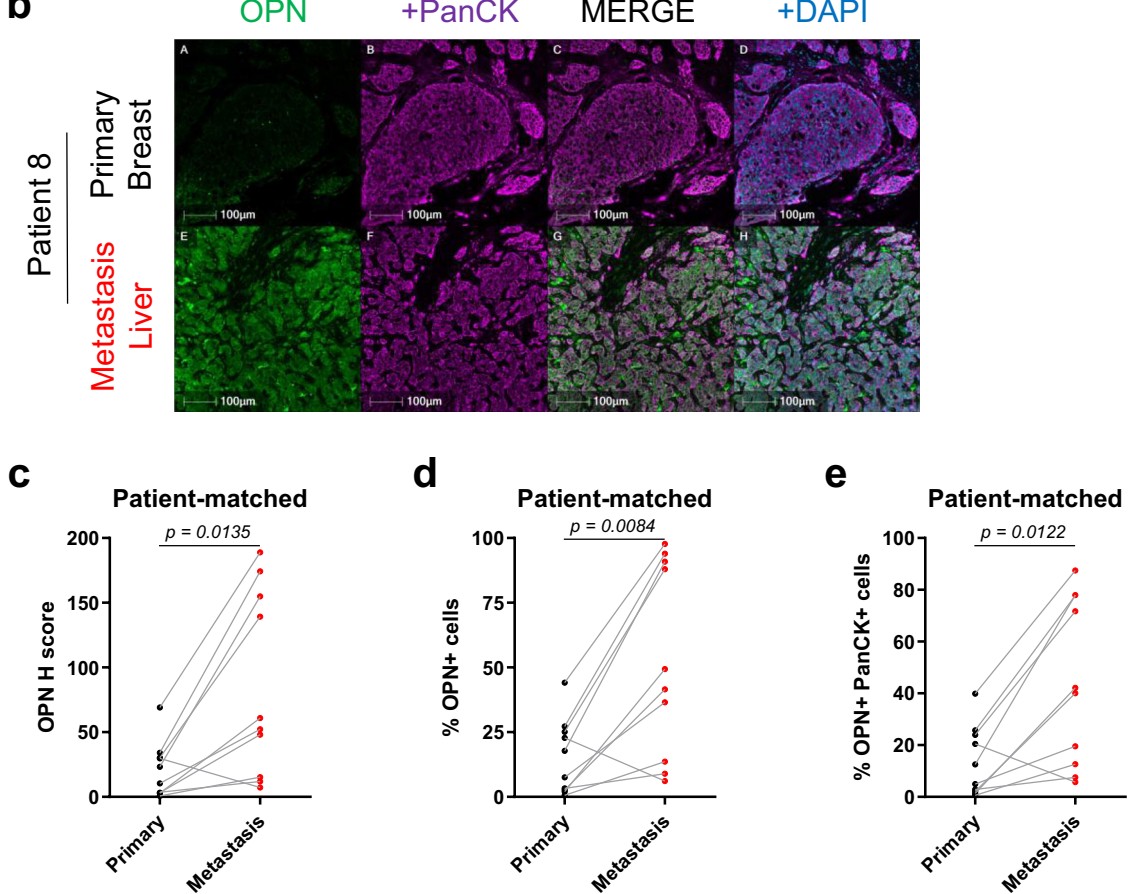

**Fig. 8 | Osteopontin levels are increased in patient-matched recurrent metastatic tumors versus primary breast tumors. a** Sample manifest of patient-matched primary breast tumors and recurrent metastatic tumors. **b** Fluorescent IHC for OPN, PanCK, and DAPI on patient-matched primary breast tumors ($n = 10$) and recurrent metastatic tumors ($n = 10$). **c**–**e** Quantification of OPN H Score, total OPN+ cells, and OPN+ PanCK+ cells in patient-matched primary breast tumors ($n = 10$) and recurrent metastatic tumors ($n = 10$). Scale bars are as indicated on each image. Mean ± SEM for data calculated using two-tailed Student's $t$ test. Each data point is representative of one biological sample for (**c**–**e**). Source data are provided as a Source Data file.

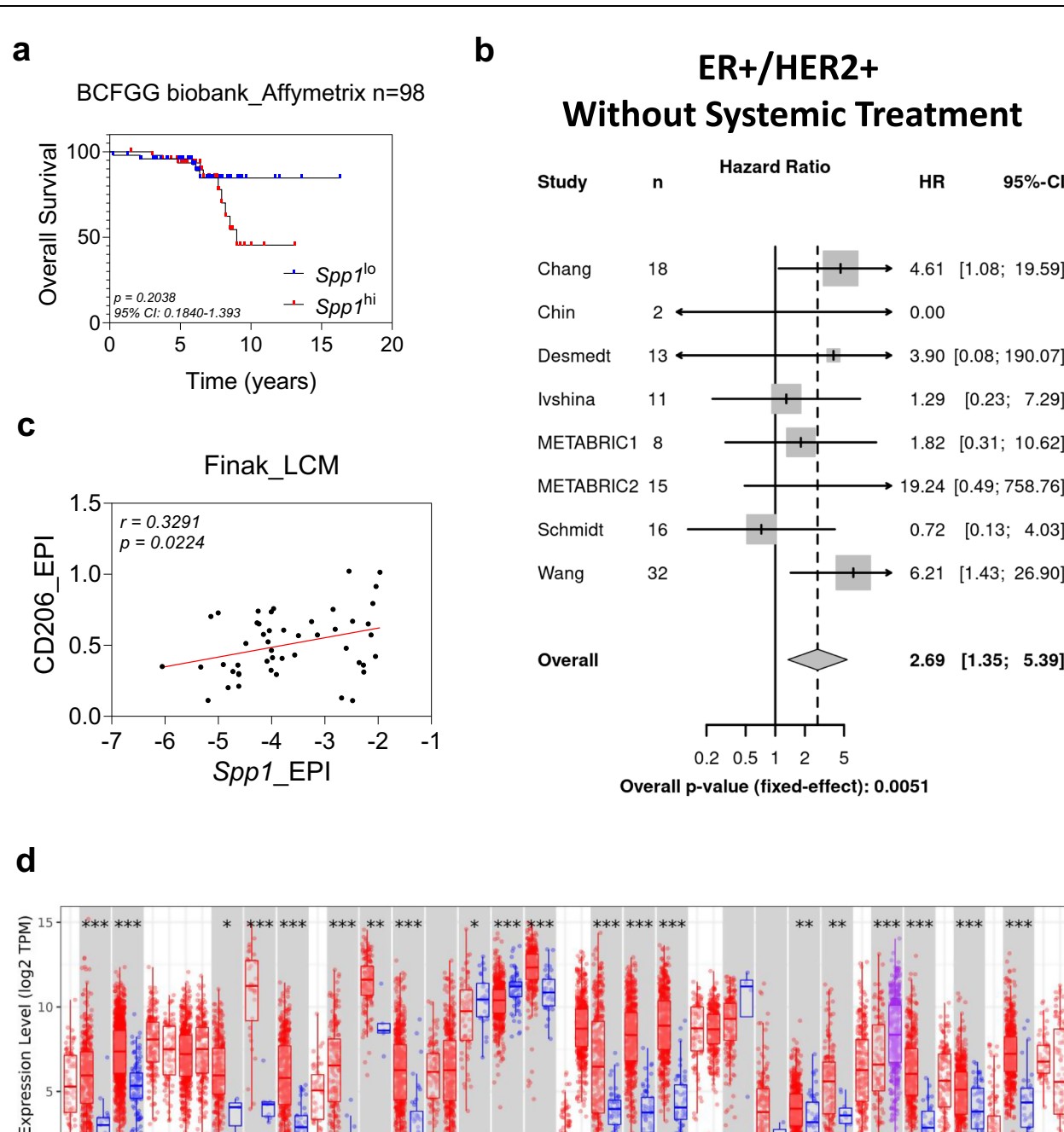

10 mM KHCO$_3$, 0.1 mM Na$_2$EDTA, pH 7.5) to remove red blood cells for 1 min at room temperature. 5 million cells were plated in each 10 cm tissue culture dish in 10% HI FBS BMDM media with 30 ng/mL of M-CSF (Peprotech 315-02). BMDM cells were supplemented with 30 ng/mL of M-CSF on day 3 and day 6. On day 6, BMDM cells were supplemented with saline, 500 ng/mL of rmOPN (R&D Systems 441-OP) and/or 20 ng/mL of IL-4 (Peprotech 214-14) for polarization for 48 h. BMDM cells were harvested by scraping for FACS.

**Transwell migration assay on BMDM cells**

BMDM cells were extracted as described above and $1 \times 10^5$ cells were seeded in 5% HI FBS BMDM media onto permeable support for 24-well plate with 8.0 μm transparent PET membrane cell culture inserts (Falcon, 353097). The inserts were then placed in 24-well plates containing 10% HI FBS BMDM media supplemented with saline, 1 or 5 μg/mL of rmOPN (R&D Systems 441-OP) for 4, 8 or 24 h of incubation at 37 °C, 5% CO$_2$. Each condition was run in triplicate. Following 4, 8 or 24 h of

**Fig. 9 | Osteopontin positively correlates with macrophage infiltration and decreased relapse-free survival in human breast cancer. a** Kaplan−Meier survival curves of *Spp1* low and *Spp1* high-expressing ER+ /PR+ /HER2- human breast tumors from GSE58644. Statistical significance was calculated using the Log-rank (Mantel-Cox) test. **b** Forest plot representation of meta-analysis on hazard ratios for 5-year relapse-free survival as a function of OPN pathway activity (*Spp1, Timp3, Col11a1, Mmp9, Mmp2, Fn1, Cd44, IL-4*) for 115 breast cancer patients across 8 individual datasets using a tailored OPN pathway activity signature. Names and sizes of data sets, HR (center of square), and 95% CIs (horizontal line) are shown for each dataset. Sizes of squares are proportional to weights used in meta-analysis. The overall HRs (dashed vertical lines) and associated CIs (lateral tips of diamond) are shown for the random-effects model. Solid vertical line indicates no effect. The HRs represent the change in risk over half of the full range of estimated pathway activity. The overall *P* value was calculated using two-tailed z-test on the pooled hazard ratio estimate. **c** Correlation between *Spp1* and CD206 in the epithelial compartment of human breast cancer from GSE9014 (*n* = 48). Statistical significance was calculated using two-tailed Spearman rank correlation test. **d** *Spp1* expression levels between tumor and adjacent normal tissues across various cancers, generated using TIMER2.0. Box plot with center line = median, box = 25th−75th quartile, whiskers = maxima/minima. Statistical significance was calculated using two-tailed Wilcoxon test (*\**p* < 0.05; \*\**p* < 0.01; \*\*\**p* < 0.001). Source data are provided as a Source Data file.

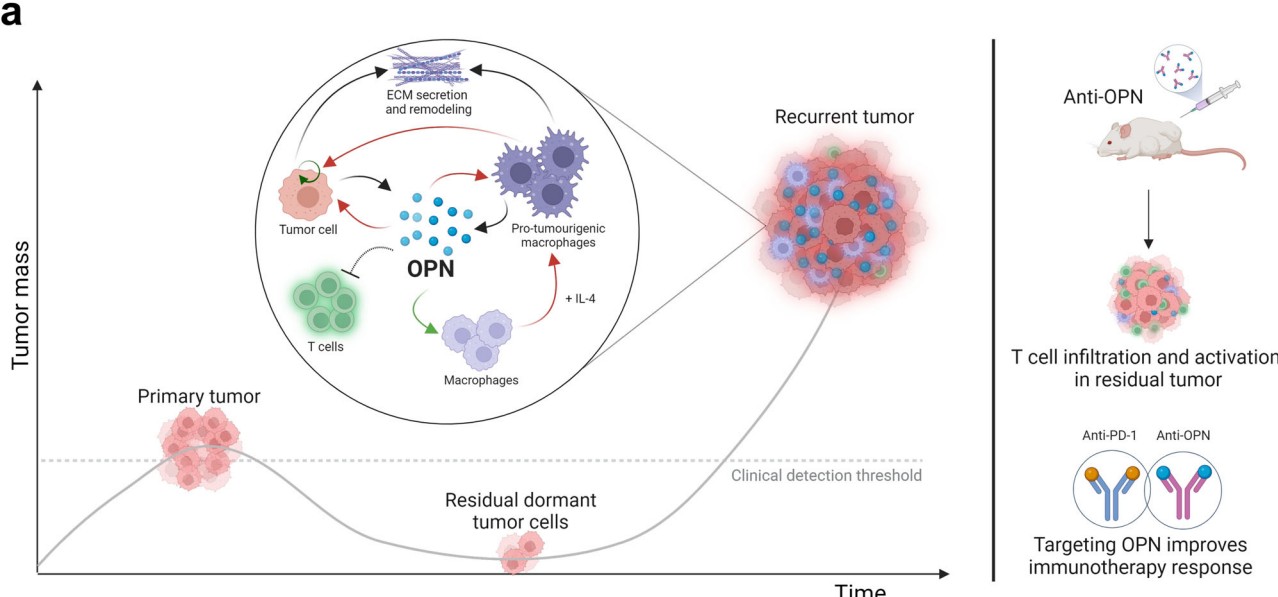

**Fig. 10 | Osteopontin is a therapeutic target that drives breast cancer recurrence. a** Schematic model of breast cancer recurrence driven by osteopontin (OPN). In the transgenic mouse model of breast cancer recurrence post-β1 integrin inhibition and in local and metastatic recurrent patient tumor samples, elevated levels of OPN serve as a central modulator that directs a pro-tumorigenic environment. OPN promotes tumor cell proliferation, recruits macrophages, synergizes with IL-4 to polarize them into the pro-tumorigenic state, and inhibits T cell activity. Tumor cell and pro-tumorigenic macrophages contribute to extracellular matrix (ECM) secretion and remodeling, including OPN. Targeting OPN reduces tumor burden, permits T cell infiltration and activation, and improves anti-PD-1 response. Black arrow indicates secretion; green arrow indicates recruitment; red arrow indicates activation; circular arrow indicates proliferation; blunt arrow indicates inhibition. Created in BioRender. Muller, W. (2024) BioRender.com/g26d472.

incubation, the filters were removed, and the non-migrating cells from the top of the membrane were wiped away with a cotton swab. The cells that migrated through the membrane were fixed in 10% neutral buffed formalin for 20 min and counterstained in 0.1% crystal violet and 20% methanol solution. Image acquisition was done on ZEISS Axio Zoom.V16 microscope (objective 30X). Cells in one representative field per image were counted manually in Fig. 4c.

### Flow cytometry
Mouse BMDM cells were fixed using BD Cytofix/CytopermTM Plus (BD Bioscience, #555028) according to manufacturer's protocol, Fixed cells were stained for viability using the fixable viability dye eFluor™ 506 (ThermoFisher/eBioscience 65-0866-14, 1:200), F4/80 (BD bioscience 123114, 1:600), CD11b (Biolegend 563168, 1:300), CD206 (Biolegend 141706, 1:200), IL-4R (Biolegend 504117, 1:400) and Arginase 1 (Invitrogen 46-3697-82, 1:400). BD LSR Fortessa flow cytometer and Flowjo 10.6.2 were used for data collection and analysis, respectively.

### TIMER 2.0 analyses
The expression of *Spp1* in solid tumor samples versus normal tissue of various cancers was generated using the "Cancer Exploration" module on TIMER2.0 (http://timer.cistrome.org/).

### Statistical analysis
All statistical analyses were done using GraphPad Prism 9.0 software. Significance between two sets of data was assessed using two-tailed Students' *t* test. Data represent mean ± SEM (standard error of the mean) for biological replication. Significance and mean ± SEM between more than two sets of data were calculated using Ordinary One Way ANOVA with Tukey's post hoc test. For Kaplan−Meier survival analysis, statistical significance was calculated by Lox-rank (Mantel-Cox) test. For cell proliferation curve, statistical significance was calculated using two-way ANOVA test with Tukey's post hoc test. For tumor kinetic growth curves and animal weight curves two-tailed Students' *t* test was performed at end point to compare two groups. Figure 9b was generated as previously described[69]. *p* < 0.05 are considered significant.

### Reporting summary
Further information on research design is available in the Nature Portfolio Reporting Summary linked to this article.

### Data availability
Single-cell RNA sequencing data and bulk RNA sequencing data that support the findings of this study have been deposited in NCBI GEO with the accession codes GSE186118 and GSE186491, respectively. Human primary breast cancer datasets that support the findings are

publicly available and obtained from NCBI GEO with the accession codes GSE58644 and GSE9014. The remaining data are available in the Article, Supplementary Information and Source Data file. Source Data provided with this paper. Source data are provided with this paper.

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

## Acknowledgements

We thank the Scientific Platforms of the Rosalind and Morris Goodman Cancer Institute, including the Histology Innovation Platform, Flow Cytometry Core Facility, Bioinformatics Core Technology Platform, and Jade Desjardins and Mitra Cowen from McGill Integrated Core for Animal Modeling. This study is funded by Canada Research Chair in Molecular Oncology, CIHR Foundation (Grant # FDN-148373 and #PLL – 190347), CCSRI (Grant # 706679 and #706216), and CCS (Grant #708195) (W.J.M.), Fonds de Recherche Québec Santé and the Canadian Institutes of Health Research (Funding Reference Number 187660) (Y.G.).

## Author contributions

Conceptualization and investigation: Y.G., W.J.M. Experimental design: Y.G. Methodology and sample acquisition: Y.G., T.T., T.B., D.Z., A.Po., S.A., V.S.G., V.P., N.U.L., M.E.H., K.S., M.P., M.L.T., and R.J. Data curation and acquisition: Y.G., T.T., D.Z., and V.S.G. Data analysis: Y.G., T.T., A.Pa., A-M.F., V.S.G., T.C.P., and L.A.C. Writing (original draft, review, and editing): Y.G. and W.J.M. Visualization: Y.G. Funding acquisition: Y.G., T.T., and W.J.M. Study supervision: W.J.M.

## Competing interests

The authors declare no competing interest.
