## [Peer Review File · Nature Communications]

Osteopontin is a therapeutic target that drives breast cancer recurrenceREVIEWER COMMENTS

Reviewer #1 (Remarks to the Author): with expertise in cancer immunology, osteopontin

The authors report the use of the MIC β 1KO mouse as a recurrent breast cancer mouse model to investigate the TME. They observed that OPN expression level is elevated in the proliferating tumor cells. They then investigated the link between β 1 KO, STAT3 activation, and OPN expression in both mouse tumor in vivo and tumor cell lines in vitro. The recurrent tumors have high STAT3 activation, higher OPN level, and increased F4/80+CD206+ macrophage tumor infiltration/recruitment. In addition, IL4 is elevated in PanCK+ cells. The author further determined that in both MIC WT and MIC β 1KO mice, % OPN+ PanCK+ epithelial cells outnumbered OPN+ F4/80+ macrophages, indicating that proliferating tumor cells are the main source of IL4 that recruit macrophages. In addition, OPN receptors are also significantly increased in MIC β 1 KO mice as compared to MIC WT mice. Transwell assays validated that OPN promotes macrophage (BMDM) migration and OPN and IL4 act together to polarize macrophages in vitro. Recombinant OPN protein therapy in MIC WT mice promotes tumor cell proliferation, increased CAF, and to a less degree increased macrophage tumor infiltration. OPN protein also promotes tumor growth in MIC β 1KO mice. As a complementary study, the author determined that OPN neutralization decreased tumor growth and lung metastasis. Although T cells are not significantly different between MIC WT and MIC β 1 KO mice despite different OPN levels in these mice, OPN neutralization significantly increased T cell activation and tumor infiltration. Analysis of human cancer patient datasets validated that OPN is elevated in high grade invasive breast cancer, correlated with macrophage infiltration in recurrent breast tumor and poor prognosis.

The significance of this manuscript is the role of OPN in breast cancer recurrence and the use of the MIC β 1KO mice that resembles human breast cancer recurrence.

The major weakness is that the novelty is not clearly stated and the lack of a clear/specific mechanism of action of OPN. The STAT3-OPN axis, the IL4-macrophage polarization axis, and the effect of OPN on T cells are known in the literature.

Comments

Suppl Fig. 1: pSTAT3 blot has more than one band. How specific is the antibody? The difference is minimal between the 2 groups. The same applies to STAT3.

Suppl Fig. 2: A: 1) β 1 integrin KO is not clear, which bands are β 1 integrin; 2) Both pSTAT3 and STAT3 levels are not obviously different from the Western blot between the 2 groups of cells. Cell line # 3 even shows lower pSTAT3 in KO cells. Yet, OPN levels are higher in all 4 KO cell lines (D).

Fig. 2D-H: IF and scRNA-seq were used to identify IL4-expressing cells, and a spatial localization of IL4, PanCK and Ki67 is shown. To further strengthen this finding, flow cytometry should be used to show the IL4+, PanCK+, and Ki67+ cell. IL4 is cytokine that is secreted protein. Flow cytometry can show the double-triple positive cells. This will strengthen the conclusion that proliferating tumor cells are the IL4+ cells.

Figure 4. OPN receptors are also significantly increased in macrophages in MIV β KO mice. The authors therefore conclude that “which align with our hypothesis that the macrophages in recurrent tumors are responsive to the tumor cell-secreted OPN”. This statement is confusing. OPN binds to its receptors to activate the downstream signaling pathways. Does OPN increase its own receptor expression? What is responsible for the tumor recurrence? OPN or OPN receptors, or both?

Figure 3D: The flow chart: there is only one cell population. No negative vs positive populations. The difference is the level. The gated cells are basically IL4R high and Arginase high cells, not IL4R+ Arginase+ cells.

The external recombinant OPN protein treatment experiments are not physiologically relevant: 1) OPN protein level is already high in both WT and KO mice (Suppl Fig. 2D), the rationale to add recombinant OPN protein to the mice is not clear; 2) as the authors pointed out, OPN is a heavily post-translationally modified protein, the external recombinant protein may be different from the endogenous OPN in the mice; and 3) The treatment dose is 5 μ g OPN protein /mouse and the mice were treated once a week. The stability of the external OPN protein in mice is not known. The endogenous OPN protein is continuously produced

by tumor cells and therefore should outnumber the external OPN. It is important to measure the half-life or other factor of the external OPN protein to determine its relevance.

A graphic summary of all the findings in the context of OPN regulation of tumor growth and recurrence will help the readers to understand the findings.

Reviewer #2 (Remarks to the Author): with expertise in breast cancer, TME, macrophages

The authors previously established a genetically engineered mouse model, known as MIC mouse, where Polyoma middle T antigen and Cre recombinase are induced in mammary epithelial cells upon doxycycline treatment. Subsequently, they introduced $\beta 1$ integrin floxed alleles into MIC mice to generate MIC $\beta 1$ KO mice, where $\beta 1$ integrin is depleted in mammary epithelial cells. Earlier observations from this model indicated that $\beta 1$ integrin depletion inhibited mammary tumor development. However, despite this suppression, 70% of MIC $\beta 1$ KO mice displayed recurrent mammary tumor development following a variable dormancy period. Here, the authors conducted single-cell RNA sequencing using end-stage mammary tumors from these unique models and revealed that recurrent tumors in MIC $\beta 1$ KO mice exhibited elevated levels of osteopontin (OPN) mRNA compared to tumors in MIC mice. Multiplex immunostaining of these tumor samples demonstrated increased expression of OPN protein and a higher abundance of F4/80+CD206+ macrophages in recurrent tumors within MIC $\beta 1$ KO mice. Moreover, their data indicated a higher percentage of proliferating tumor cells adjacent to F4/80+CD206+ macrophages in MIC $\beta 1$ KO recurrent tumors compared to MIC WT tumors. Additionally, the authors found a positive correlation between OPN expression, macrophage infiltration, and decreased relapse-free survival in human breast cancer. These findings provide novel insights into the tumor microenvironment associated with breast cancer recurrence.

The roles of OPN in promoting breast cancer development has been reported in several studies. For example, OPN expression is increased both in situ and invasive breast carcinoma lesions compared to normal breast tissues (Am J Pathol 1995, 146:95-100). In mouse models, ectopic expression of OPN in human breast cancer cells promotes in vivo tumor

formation, whereas OPN knockdown results in reduced tumor growth (Clin Exp Metastasis 2006, 23:123-133; Cell Mol Immunol 2013, 10: 176–182; Oncogene 2014, 33: 2053–2064). It is also demonstrated that the treatment with anti-OPN antibody can suppress primary tumor growth and spontaneous metastasis (Cancer Immunol Immunother 2010, 59:355–366). Furthermore, OPN knockout in host cells has been reported to increase pulmonary metastasis of cancer cells, coupled with CD4+ T cell activation at the metastatic site (Cancer Res 2014, 74: 4706–4719). Additionally, there are some studies suggesting that OPN promotes the migration and alternative M2 activation of TAMs (reviewed in Front Oncol. 2022, 12:953283). The results presented in this manuscript on MIC WT tumors, in vitro assays, and human invasive breast cancer samples are consistent with and support the results of previous studies.

This study is distinguishable from previous research by its specific focus on the tumor microenvironment, particularly macrophages therein, in the recurrence of breast cancer. Nevertheless, there are areas for improvement to substantiate the authors' claims. The primary concerns and comments for each figure are outlined below.

Primary concerns

- 1) There is currently insufficient evidence to suggest that the increase in OPN is necessary for the recurrence of mammary tumors in MIC β 1KO mice.
 - The administration of recombinant OPN did not lead to a shortened dormancy period for tumors in MIC β 1KO mice. While the authors stated that the mammary glands of MIC β 1KO mice already exhibit significantly higher levels of OPN compared to MIC WT mice, it is important to note that data in Fig. 1 were obtained from tumors at the experimental endpoint. Thus, it remains unclear whether OPN expression is increased in early-stage tumors in MIC β 1KO mice compared to that in MIC WT mice. Investigating the kinetics of OPN expression in MIC WT and MIC β 1KO tumors, as well as the OPN level within MIC β 1KO tumors showing stable dormant growth at the endpoint, could elucidate a potential correlation between OPN expression and tumor recurrence.
 - Exploring the impact of the OPN neutralizing antibody on tumor development in MIC β 1KO mice, alongside MIC WT mice, would provide further insights.
 - It is worth noting that a study has reported that tumor incidence, growth rate, or

macrophage accumulation are not affected by OPN knockout in another mammary tumor mouse model, MMTV-c-myc:Ha-ras (Breast Cancer Res Treat 2000, 63:71–79). Referencing this paper would enrich the discussion.

2) Further exploration into the characterization of tumor-associated macrophages (TAMs) could provide deeper insights into the tumor microenvironment associated with the tumor recurrence.

- Conducting additional analysis of single-cell RNA sequencing data can unveil distinct macrophage subpopulations within MIC WT tumors and MIC β 1KO recurrent tumors, along with their corresponding differentially expressed genes. These analyses can unveil representative markers and potential origins of each TAM subset, which will provide a rationale for directing attention towards CD206+ monocyte-derived macrophages in subsequent experiments.

- The impact of recombinant OPN on tumor cell proliferation (Fig. S2) appears modest, potentially supporting the author's hypothesis regarding the involvement of TAMs in recurrent tumor outgrowth. This hypothesis could be supported through an investigation into the effects of TAMs isolated from MIC β 1KO recurrent tumors on tumor cell proliferation and apoptosis.

- Data in Fig. 2A-C cannot exclude the possibility that the increase in CD206+ TAMs simply reflects an increase in the total number of macrophages or other cytokines are involving in the expression of CD206. Therefore, the author's statement "IL-4 to polarize macrophages for a favorable TME in MIC β 1KO recurrent tumors" is not fully supported by the current data.

- The data presented in Fig. S4 do not rule out the possibility that there was an increase in a TAM subpopulation inherently expressing OPN receptors. Additionally, besides CD44 and α V β 3 integrin, other integrins such as α V β 1, α V β 5, α V β 6, α 4 β 1, α 5 β 1, and α 9 β 1 are recognized as OPN receptors. Conducting multi-color flow cytometry will identify whether specific TAM subsets (e.g., CD206+ macrophages) co-express these receptors.

- While authors state that recombinant OPN increased BMDM polarization in vitro, recombinant OPN injection did not increase the number of CD206+ TAMs in vivo. Does OPN affect the expression of CD206 in BMDM? Are the Arginase1+IL-4 receptor+ macrophages

increased in the recurrent tumors? Does OPN have no effect on the phenotype or function of tissue-resident macrophages?

Other comments on each figure:

Fig. 1F,G: The morphology of MIC β 1KO recurrent tumors appears distinct from that of MIC WT tumors. Further clarification regarding the histological tumor grade of each would be beneficial for interpretation.

Fig. S1H-J: Does MIC Stat3 knockout exhibit a similar effect on tumor growth compared to MIC β 1KO?

Fig. S2A: The interpretation of the immunoblot for β 1 integrin is challenging, raising concerns about the efficacy of β 1 integrin knockout. Evaluation via flow cytometry would provide more convincing data.

Fig. S2E,F: Given the study's focus, it is crucial to investigate the effects of OPN on the proliferation of MIC β 1KO tumor cells.

Fig. 2G: Enhancing the visibility of Beige dots by changing their color and providing higher magnification images would aid readers in data assessment. Clarifying the reference point for the 'percentage of PanCK+ Ki67+ cells' would enhance data understanding. Additionally, comparing the percentage of Ki67+PanCK+ cells within total PanCK+ cells located $<50\mu\text{m}$ from macrophages to that within total PanCK+ cells located $>50\mu\text{m}$ away would offer a comprehensive perspective.

Fig. 4, Fig. 5: Please clarify the reference point for the percentage values.

Fig. S8A,B: The description of the data is difficult to comprehend. Clarification would improve understanding.

Fig. 8, Fig S10: While I understand the challenges in accessing clinical samples, comparing primary and metastatic tumors may not be suitable for this study. Instead, focusing on local recurrence tumors would provide a more relevant comparison with the mouse data.

Reviewer #3 (Remarks to the Author): with expertise in breast cancer

The manuscript by Gu et al investigates the effect of osteopontin on primary breast tumor progression and metastatic recurrence. The authors use a unique model (MIC WT and MIC β 1KO) to explore how OPN influence tumor proliferation and macrophage infiltration. The

idea that OPN influences tumor progression is not novel, but the data are interesting and the careful quantification of images throughout is appreciated. However, there are issues that need to be addressed:

Major issues:

1. At the beginning, it is not clear why the integrin B1 KO was used / introduced. This becomes more confusing about halfway through the results with the sentence “It is well established that OPN can stimulate cell migration, mainly by engaging with integrin receptors [7].” Is the idea that integrin B3 is regulating OPN, or that when integrin B3 is knocked out OPN cannot signal through it anymore? Since many of the experiments presented are on the MIC B1 KO background, the rationale for its use at the beginning needs to be better explained.
2. The data suggesting OPN increases tumor proliferation through an autocrine or paracrine effect is not convincing. Supp2E does not match with the statistics – it is showing a highly significant effect of rmOPN on proliferation but appears to not change confluency based on the y-axis scale. In 2F OPN also does not appear to have much effect on proliferation. Later data in Figure 4 are more convincing that OPN increases proliferation with the Ki67 staining, but the effect is not consistent throughout the manuscript.
3. The main hypothesis seems to be that OPN is secreted by the tumor cells, which increases recruitment of macrophages with OPN receptors, polarizes them to be pro-tumorigenic, which then supports tumor growth and exit from dormancy. However, there are no measures of dormancy or proliferation (Ki67, mitotic index, p27, p16, etc) in the tumor cells in the early studies (again these come up in Figure 4, but the earlier data is weaker by not including these markers). Would it not be more direct to knockdown the elevated OPN observed in the B1 KO and show that this reverses the increase in OPN receptor+ macrophages?
4. The effect of a-OPN on the tumors seems to drive the reduction in tumor growth with a-OPN + a-PD-1. This does not necessarily support the idea that a-OPN re-sensitizes the tumors to a-PD-1 since they do not appear to grow slower than a-OPN treatment alone. The authors suggest that targeting OPN may be a “viable therapeutic strategy to improve

immunotherapy response, but they have not demonstrated this. This conclusion would need to be supported by data demonstrating that there is increased T-cell-mediated killing of the tumor cells in vivo when a-OPN and a-PD-1 are combined, and the effect would need to be greater than that of a-PD-1 alone and a-OPN alone. An increase in T cell infiltration is not sufficient to indicate that T cell-mediated killing is happening (e.g., they may be exhausted).

5. The finding that T cell infiltration increases with OPN treatment is interesting and striking in the images. Does PD-L1 expression in the tumor change? Understanding the status of PD-L1 is important for confirming the mechanism. Are the T cells functional, or do they express exhaustion markers (e.g. Tim3, Lag3)?

6. The in vivo data does not directly demonstrate the role of OPN in tumor recurrence, so need to tone down statements like “in perfect concordance with our in vivo data” in reference to the patient data in Figure 8. In Figure 8B, it is difficult to determine how much PanCK+ tumor overlaps with OPN. Please add a panel of PanCK+ cells only (not merged with OPN) and quantitation of the % OPN+ PanCK+ cells as have been done for other figures. While Fig 8D&E are supportive, the correlation of CD68+ cells with OPN+ cells is quite weak (as is 9C), and does not fit with the lack of a change in CD68+ macrophages in the metastatic vs primary tumor samples (Fig 8C). The patient data strongly support a potential role for OPN in metastatic recurrence, but not through an increase in macrophage infiltration.

7. The authors proposed working model does not fit with the proposed model from Weinberg in colleagues in 2008, where they proposed that OPN is secreted by the primary tumor and that promotes recurrence in the secondary site by mobilizing cells in the bone marrow. The patient data presented herein demonstrate that OPN is expressed at higher levels in the metastatic site. Please include more on this in the discussion and place the findings in the context of the field (they do not have to agree but need to be discussed more in depth to provide context).

8. The conclusions are strongly stated and sometimes overstated throughout the results and discussion. For example, “Collectively, these data argue that increased OPN secretion by mammary epithelial tumor cells within recurrent tumors provides the necessary tumor cell

autonomous and microenvironmental support for cancer recurrence. This includes restructuring an immune and fibrotic TME landscape through OPN-responsive pro-tumorigenic macrophages” – the authors have not conclusively demonstrated this. There are several strong statements with regard to the patient data validating the in vivo data, but these do not align perfectly and should be toned down. For example, “Unquestionably, our extensive patient data underscore the relevance and accuracy of our findings in vivo.”

Minor issues:

9. Please clarify in the methods whether the MIC Stat3 KO mouse is a global KO or a mammary epithelium specific Stat3 KO. Based on the text I believe it was mammary epithelium-specific, but could be clarified.
10. Western blots in Supp 2A are not convincing for increased pStat3 or Stat3, and do not appear to match with panel B.
11. The IL-4/PanCK co-localization is difficult to see. Can the channels be two more distinct colors? They are too similar in color to appreciate co-localization.
12. Figure 2 G-H could also indicate that the tumor cells secrete more chemoattractants for the F4/80+ macrophages. Is MCP-1 elevated?
13. I do not think autocrinally is a word.
14. CD44 does co-localize with F4/80+ cells, but presumably also with PanCK+ tumor cells, which are missing from Supp 4E.
15. Supp 5B is too small to read.
16. The quantification for Supp 7B does not match the image in panel A.
17. Please clarify the findings of Fig 9B in the text – the figure panel is confusingly presented.
18. Define EMC in the discussion.
19. In the methods mouse tissue collection describes bone samples that were collected and decalcified for histomorphometry but these data are not included. Are there bone metastasis samples that are available and could be reported?
20. Please include more detail under human samples in the methods on how they were stained.

Reviewer #4 (Remarks to the Author): with expertise in cancer omics

(1) “we used CellPhoneDB to visualize all cell clusters that express OPN receptor complexes in response to epithelial cell (Cluster 0)-secreted OPN [17]. ...” Recent studies have shown that CellPhoneDB and CellChat have very low specificity in detection of cell 2 cell communications (<https://www.biorxiv.org/content/10.1101/2023.09.18.558298v4>). I would suggest to remove these results.

(2) “We further analyzed Kaplan-Meier survival curves generated from dataset GSE58644, ...” Please also report the HR+95% confidence interval in addition to p value

Dear Reviewers,

We sincerely thank all four Reviewers for your insightful comments and feedback on our manuscript entitled ‘*Tumor-derived osteopontin is a therapeutic target that drives breast cancer recurrence.*’ We have edited the manuscript to address your concerns, and here, would like to thoroughly clarify on each of your comments.

REVIEWER #1

- 1. Suppl Fig. 1: pSTAT3 blot has more than one band. How specific is the antibody? The difference is minimal between the 2 groups. The same applies to STAT3.**
 - a. The antibody for p-Stat3 is specifically for Tyr705 and does normally show two distinct bands on immunoblots at 79 and 86 kDa (reference: Cell Signaling Technology Phospho-Stat3 (Tyr705) (D3A7) XP® Rabbit mAb #9145). It is a widely used and accepted antibody to detect p-Stat3. The same applied for total Stat3 (reference: Cell Signaling Technology Stat3 (124H6) Mouse mAb #9139), specifically, detecting two bands at 79 and 86 kDa.
- 2. Suppl Fig. 2: A: 1) β 1 integrin KO is not clear, which bands are β 1 integrin; 2) Both pSTAT3 and STAT3 levels are not obviously different from the Western blot between the 2 groups of cells. Cell line # 3 even shows lower pSTA3 in KO cells. Yet, OPN levels are higher in all 4 KO cell lines (D).**
 - a. We have re-run the immunoblot for β 1 integrin to ensure the clarity of the blot. The β 1 integrin band is around 130 kDa. Additionally, we do agree that the total Stat3 and the p-Stat3 bands are not as obviously strong in the AdCre MMTV-PyV mT *Itgb1* fl/fl cell lines by eye, however, we did normalize each band to their respective control (β -actin), the ratio of which is quantified. Cell line #3 corresponds to the minimally increased OPN secretion (Supplementary Fig. 2d) between all 4 lines.
- 3. Fig. 2D-H: IF and scRNA-seq were used to identify IL4-expressing cells, and a spatial localization of IL4, PanCK and Ki67 is shown. To further strengthen this finding, flow cytometry should be used to show the IL4+, PanCK+, and Ki67+ cell. IL4 is cytokine that is secreted protein. Flow cytometry can show the double-triple positive cells. This will strengthen the conclusion that proliferating tumor cells are the IL4+ cells.**
 - a. We would like to kindly clarify that Fig. 3g-I (previously Fig. 2g-h) only show the spatial relationship between pro-tumorigenic macrophage (F4/80+ CD206+ cells) and proliferating epithelial cells (PanCK+ Ki67+ cells), without consideration for their secretion of IL-4. Separately, Fig. 3d-f (previously Fig. 2d-f) demonstrates that most IL-4 co-localize with PanCK, showing their epithelial origin using RNA Scope in situ hybridization instead of sequencing. However, although we currently do not possess any fresh recurrent β 1KO tumors that will allow us to perform FACS

analysis, we fully agree that proliferating and IL-4-secreting epithelial cells will further support that the same cells exiting dormancy are simultaneously polarizing macrophages to support recurrence.

4. Figure 4. OPN receptors are also significantly increased in macrophages in MIV β KO mice. The authors therefore conclude that “which align with our hypothesis that the macrophages in recurrent tumors are responsive to the tumor cell-secreted OPN”. This statement is confusing. OPN binds to its receptors to activate the downstream signaling pathways. Does OPN increase its own receptor expression? What is responsible for the tumor recurrence? OPN or OPN receptors, or both?

- a. We fully agree with both statements provided by Reviewer #1. Both OPN (in our model mainly secreted by epithelial cells) and OPN receptors (can be expressed on all cells within a tumor) are contributing to recurrence. Our new data demonstrate that OPN acts on macrophages to promote tumor recurrence. Targeting OPN or depleting macrophages inhibited tumor outgrowth in the MIC β 1KO mouse model (Supplementary Fig. 7a-b and Fig. 5, respectively). The expression of OPN receptors on macrophages is vital to promote their migration and thus we believe that both OPN and its receptors are responsible for tumor recurrence (Fig. 4 and Supplementary Fig. 5). In addition, as requested by Reviewer #2, we have performed extensive fluorescent IHC for various OPN receptors (Supplementary Fig. 5e-j) and demonstrated their expression on CD206+ macrophages in the MIC β 1KO recurrent tumors. While the exact molecular pathways activated in macrophages by OPN are outside the scope of this study, our *in vitro* data on BMDMs demonstrate that OPN stimulates macrophage migration (Fig. 4a-c). Several studies have elucidated the different pathways activated by OPN, these include Stat3 pathway as well as integrin activated pathways (1-4). We have also added a brief section about pathways activated by OPN in the revised manuscript.
- b. Reference 1: Lund SA, Wilson CL, Raines EW, Tang J, Giachelli CM, Scatena M. Osteopontin mediates macrophage chemotaxis via α 4 and α 9 integrins and survival via the α 4 integrin. *J Cell Biochem.* 2013 May;114(5):1194-202. doi: 10.1002/jcb.24462. PMID: 23192608.
- c. Reference 2: Wei J, Marisetty A, Schrand B, Gabrusiewicz K, Hashimoto Y, Ott M, Grami Z, Kong LY, Ling X, Caruso H, Zhou S, Wang YA, Fuller GN, Huse J, Gilboa E, Kang N, Huang X, Verhaak R, Li S, Heimberger AB. Osteopontin mediates glioblastoma-associated macrophage infiltration and is a potential therapeutic target. *J Clin Invest.* 2019 Jan 2;129(1):137-149. doi: 10.1172/JCI121266. Epub 2018 Nov 19. PMID: 30307407; PMCID: PMC6307970.
- d. Reference 3: Choi SI, Kim SY, Lee JH, Kim JY, Cho EW, Kim IG. Osteopontin production by TM4SF4 signaling drives a positive feedback autocrine loop with the STAT3 pathway to maintain cancer stem cell-like properties in lung cancer cells.

Oncotarget. 2017 Sep 18;8(60):101284-101297. doi: 10.18632/oncotarget.21021. PMID: 29254164; PMCID: PMC5731874.

- e. Reference 4: Wooten DK, Xie X, Bartos D, Busche RA, Longmore GD, Watowich SS. Cytokine signaling through Stat3 activates integrins, promotes adhesion, and induces growth arrest in the myeloid cell line 32D. *J Biol Chem*. 2000 Aug 25;275(34):26566-75. doi: 10.1074/jbc.M003495200. PMID: 10858439; PMCID: PMC2396147.
5. **Figure 3D: The flow chart: there is only one cell population. No negative vs positive populations. The difference is the level. The gated cells are basically IL4R high and Arginase high cells, not IL4R+ Arginase+ cells.**
 - a. We agree with Reviewer #1 and have adjusted the Y axis titles and figure legend to Arg1^{high} and IL4-R^{high}, instead of Arg1⁺ and IL4-R⁺, respectively.
6. **The external recombinant OPN protein treatment experiments are not physiologically relevant: 1) OPN protein level is already high in both WT and KO mice (Suppl Fig. 2D), the rationale to add recombinant OPN protein to the mice is not clear; 2) as the authors pointed out, OPN is a heavily post-translationally modified protein, the external recombinant protein may be different from the endogenous OPN in the mice; and 3) The treatment dose is 5 ug OPN protein /mouse and the mice were treated once a week. The stability of the external OPN protein in mice is not known. The endogenous OPN protein is continuously produced by tumor cells and therefore should outnumber the external OPN. It is important to measure the half-life or other factor of the external OPN protein to determine its relevance.**
 - a. We agree with Reviewer #1 that the commercially available recombinant OPN protein lacks the biological post-translational modifications and cannot perfectly mirror OPN secreted by epithelial cancer cells *in vivo*. However, as we do see a physiological effect in the treated mice which is in concordance with our hypothesis and previously published studies that OPN accelerates tumor growth, we have decided to keep but modify Fig. 2 and Supplementary Fig. 3, previously Fig. 5 and Supplementary Fig. 6. The dosage of the treatment is based on previous studies using the same recombinant OPN protein for intraperitoneal delivery in mice (1). We agree that the endogenous OPN secreted by epithelial cancer cells, especially by MIC β 1KO epithelial cancer cells, could outweigh the exogenous OPN. To this end, we evaluated the levels of OPN in the post-treated mammary glands and tumors (Supplementary Fig. 3). In MIC WT mice, OPN level rose from ~15% OPN+ cells to ~30% OPN+ cells in rmOPN group, whereas MIC β 1KO treated and control groups remained consistently high at ~35% OPN+ cells.
 - b. Reference 1: Xanthou G, Alissafi T, Semitekolou M, Simoes DC, Economidou E, Gaga M, Lambrecht BN, Lloyd CM, Panoutsakopoulou V. Osteopontin has a crucial role in allergic airway disease through regulation of dendritic cell subsets.

Nat Med. 2007 May;13(5):570-8. doi: 10.1038/nm1580. Epub 2007 Apr 15. PMID: 17435770; PMCID: PMC3384679.

7. **A graphic summary of all the findings in the context of OPN regulation of tumor growth and recurrence will help the readers to understand the findings.**
 - a. We fully agree that a graphic summary is highly beneficial to summarize our work and did include a detailed one highlighting the main findings as Fig. 10.

REVIEWER #2

1. **There is currently insufficient evidence to suggest that the increase in OPN is necessary for the recurrence of mammary tumors in MIC β 1KO mice. The administration of recombinant OPN did not lead to a shortened dormancy period for tumors in MIC β 1KO mice. While the authors stated that the mammary glands of MIC β 1KO mice already exhibit significantly higher levels of OPN compared to MIC WT mice, it is important to note that data in Fig. 1 were obtained from tumors at the experimental endpoint. Thus, it remains unclear whether OPN expression is increased in early-stage tumors in MIC β 1KO mice compared to that in MIC WT mice. Investigating the kinetics of OPN expression in MIC WT and MIC β 1KO tumors, as well as the OPN level within MIC β 1KO tumors showing stable dormant growth at the endpoint, could elucidate a potential correlation between OPN expression and tumor recurrence.**
 - a) We fully agree with Reviewer #2's feedback that there is a lack of literature on the role of OPN in breast cancer recurrence, hence the novelty of our work. We also recognize that investigating the kinetics of OPN expression as MIC WT and MIC β 1KO mice progress from early tumor initiation, through early invasive carcinoma (dormant stage for MIC β 1KO tumors), until re-emerging tumors (recurrence) will provide further support in OPN's role during recurrence. To address this, we provided immunostainings for OPN on mammary glands at 2 week post-induction (early tumor initiation stage) and on early invasive carcinomas (dormant stage for MIC β 1KO tumors, 6 week post-induction counter part for MIC WT control) that show no significant differences between the two groups, at both timepoints (Supplementary Fig. 2g-j). This validates that dormant MIC β 1KO tumors begin upregulating OPN transcription during early invasive/dormant stage (Fig. 1a-b), but only significantly increase its translation and secretion during recurrence (Fig. 1c-g), supporting its contribution during the latter part of tumor progression. The revised manuscript has been modified to incorporate these changes accordingly.
2. **Exploring the impact of the OPN neutralizing antibody on tumor development in MIC β 1KO mice, alongside MIC WT mice, would provide further insights.**
 - a) We fully agree with Reviewer #2 that treating MIC β 1KO recurrent tumors with anti-mOPN would complement our anti-mOPN treatment in MIC WT mice and

provide further insights and validation for this study. To this end, given the slow kinetic and extremely low penetrance of MIC β 1KO mice tumor development, we have transplanted MIC β 1KO recurrent tumors into the mammary fat pad of FVB mice while on doxycycline. Following the same treatment regime once tumors arose, we observed the same significant effect of slower tumor growth in MIC β 1KO recurrence tumors treated with anti-mOPN (Supplementary Fig. 7a-b). The revised manuscript has been modified to incorporate these changes accordingly.

3. It is worth noting that a study has reported that tumor incidence, growth rate, or macrophage accumulation are not affected by OPN knockout in another mammary tumor mouse model, MMTV-c-myc:Ha-ras (Breast Cancer Res Treat 2000, 63:71–79). Referencing this paper would enrich the discussion.

- a) We have included this in our revised manuscript, emphasizing that our observations *in vivo* are dependent on the oncogene driving mammary tumorigenesis and the mouse background. For instance, the correlation between OPN and macrophage infiltration was negligible in MMTV-c-myc/MMTV-v-Ha-ras-driven mammary tumors yet prominently in our model, PyV mT-driven mammary tumor cells do secrete a unique set of immunosuppressive and chemotactic cytokines, including IL-4, that might not be reflected in other oncogene-driven tumors (1-2).
- b) Reference 1: Feng F, Rittling SR. Mammary tumor development in MMTV-c-myc/MMTV-v-Ha-ras transgenic mice is unaffected by osteopontin deficiency. *Breast Cancer Res Treat.* 2000 Sep;63(1):71-9. doi: 10.1023/a:1006466516192. PMID: 11079161.
- c) Reference 2: Attalla S, Taifour T, Bui T, Muller W. Insights from transgenic mouse models of PyMT-induced breast cancer: recapitulating human breast cancer progression *in vivo*. *Oncogene.* 2021 Jan;40(3):475-491. doi: 10.1038/s41388-020-01560-0. Epub 2020 Nov 24. PMID: 33235291; PMCID: PMC7819848.

4. Further exploration into the characterization of tumor-associated macrophages (TAMs) could provide deeper insights into the tumor microenvironment associated with the tumor recurrence. Conducting additional analysis of single-cell RNA sequencing data can unveil distinct macrophage subpopulations within MIC WT tumors and MIC β 1KO recurrent tumors, along with their corresponding differentially expressed genes. These analyses can unveil representative markers and potential origins of each TAM subset, which will provide a rationale for directing attention towards CD206+ monocyte-derived macrophages in subsequent experiments.

- a) We fully agree that additional analyses on the macrophage clusters of our single-cell RNA sequencing data would be insightful. To this end, we further performed sub-clustering analyses on the two macrophage clusters with their predicted cell types (Supplementary Fig. 6d-g). The majority of the sub-clustered cells are CD14/CD16 monocytes. We also interrogated *Mrc1* (CD206) expressions in these

two clusters, providing additional validation that MIC β 1KO tumors express more than MIC WT. The revised manuscript has been modified to incorporate these changes accordingly.

5. The impact of recombinant OPN on tumor cell proliferation (Fig. S2) appears modest, potentially supporting the author's hypothesis regarding the involvement of TAMs in recurrent tumor outgrowth. This hypothesis could be supported through an investigation into the effects of TAMs isolated from MIC β 1KO recurrent tumors on tumor cell proliferation and apoptosis.

- a) We fully agree with Reviewer #2 that isolating macrophages from MIC β 1KO recurrent tumors for subsequent co-culture with MIC β 1KO tumor cell would provide valuable insight into the pro-tumorigenic effects of these TAMs in a more isolated environment. Unfortunately, the lack of sufficient size and number of MIC β 1KO recurrent tumors, given its slow tumor kinetics and low penetrance, does not allow for their isolation by flow cytometry. Additionally, the tissue culture media for macrophages and epithelial cells are too different for prolonged co-culture. To this end, we are addressing Reviewer #2's concern by depleting macrophages in MIC β 1KO mice using clodronate-liposomes (1-3). As expected, treated mice had a significant reduction in tumor growth (Figure 5). This validates the TAM's pro-tumorigenicity in recurrent tumor outgrowth. The revised manuscript has been modified to incorporate these changes accordingly.
- b) Reference 1: <https://clodronateliposomes.com/>
- c) Reference 2: Bu L, Gao M, Qu S, Liu D. Intraperitoneal injection of clodronate liposomes eliminates visceral adipose macrophages and blocks high-fat diet-induced weight gain and development of insulin resistance. *AAPS J.* 2013 Oct;15(4):1001-11. doi: 10.1208/s12248-013-9501-7. Epub 2013 Jul 4. PMID: 23821353; PMCID: PMC3787235.
- d) Reference 3: Van Rooijen N, Sanders A. Liposome mediated depletion of macrophages: mechanism of action, preparation of liposomes and applications. *J Immunol Methods.* 1994 Sep 14;174(1-2):83-93. doi: 10.1016/0022-1759(94)90012-4. PMID: 8083541.

6. Data in Fig. 2A-C cannot exclude the possibility that the increase in CD206+ TAMs simply reflects an increase in the total number of macrophages or other cytokines are involving in the expression of CD206. Therefore, the author's statement "IL-4 to polarize macrophages for a favorable TME in MIC β 1KO recurrent tumors" is not fully supported by the current data.

- a) We agree that in addition to epithelial-derived IL-4, other cytokines can very well be present and contribute to the recruitment and/or polarization of macrophages in MIC β 1KO recurrent tumors. We have modified the statement in the revised manuscript accordingly.

7. The data presented in Fig. S4 do not rule out the possibility that there was an increase in a TAM subpopulation inherently expressing OPN receptors. Additionally, besides CD44 and $\alpha V\beta 3$ integrin, other integrins such as $\alpha V\beta 1$, $\alpha V\beta 5$, $\alpha V\beta 6$, $\alpha 4\beta 1$, $\alpha 5\beta 1$, and $\alpha 9\beta 1$ are recognized as OPN receptors. Conducting multi-color flow cytometry will identify whether specific TAM subsets (e.g., CD206+ macrophages) co-express these receptors.

a) We agree that analyzing OPN receptor expression on pro-tumorigenic macrophages, rather than in all macrophages, would provide additional specificity for this study. Therefore, we modified Supplementary Fig. 5e-j with new fluorescence IHC, including CD206 marker, and quantifications for OPN receptor-expressing pro-tumorigenic macrophages. We've also included an additional OPN receptor marker and quantification of F4/80+ CD206+ $\alpha 5$ integrin+ cells, as per Reviewer #2's suggestion. The same experiment was done for $\beta 5$ integrin, with concurrent results of significantly higher F4/80+ CD206+ $\beta 5$ integrin+ cells in MIC $\beta 1$ KO recurrent tumors (results not included in revised figure and manuscript due to space constraint, results presented here). We agree that a comprehensive FACS panel for more OPN receptor markers and macrophage subpopulations would provide more data, however, we unfortunately do not possess any fresh MIC WT and MIC $\beta 1$ KO recurrent tumors for this protocol. Nevertheless, our extensive immunostaining results support that CD206+ macrophages within recurrent tumors express more OPN receptors. The revised manuscript has been modified to incorporate these changes accordingly.

8. While authors state that recombinant OPN increased BMDM polarization in vitro, recombinant OPN injection did not increase the number of CD206+ TAMs in vivo. Does OPN affect the expression of CD206 in BMDM? Are the Arginase1+IL-4 receptor+ macrophages increased in the recurrent tumors? Does OPN have no effect on the phenotype or function of tissue-resident macrophages?

a) Supplementing recombinant osteopontin to BMDMs does not affect their CD206 expression (FACS, same protocol as for Fig. 4d-f). We further added to the revised Supplementary Fig. 3 fluorescent IHC staining for F4/80, CD206, arginase 1, and PanCK. Quantification showed that the percentage of F4/80+ CD206+ Arg1+ PanCK- cells is higher in MIC $\beta 1$ KO recurrent tumors. PanCK was used as a negative marker to prevent false-positives from channel cross talking and to

exclude Arg1+ epithelial cells. To further assess the functional effects of OPN on macrophages, RNA sequencing and subsequent validation on OPN-stimulated BMDMs would provide valuable insights. This falls outside of the scope of this particular project and will be followed up on by another researcher in our group.

9. Fig. 1F,G: The morphology of MIC β 1KO recurrent tumors appears distinct from that of MIC WT tumors. Further clarification regarding the histological tumor grade of each would be beneficial for interpretation.

- a) One of the features of MIC β 1KO recurrent tumors is indeed their distinct remodelling of acellular stromal tumor environment, with a notable increase in ECM deposition. This is due to an increase in cancer-associated fibroblasts in MIC β 1KO recurrent tumors (1). We reported and discussed these findings extensively in our first article characterizing this transgenic mouse model and clarified its features in the revised manuscript (1).
- b) Reference 1: Bui T, Gu Y, Ancot F, Sanguin-Gendreau V, Zuo D, Muller WJ. Emergence of β 1 integrin-deficient breast tumours from dormancy involves both inactivation of p53 and generation of a permissive tumour microenvironment. *Oncogene*. 2022 Jan;41(4):527-537. doi: 10.1038/s41388-021-02107-7. Epub 2021 Nov 15. PMID: 34782719; PMCID: PMC8782722.

10. Fig. S1H-J: Does MIC Stat3 knockout exhibit a similar effect on tumor growth compared to MIC β 1KO?

- a) MIC Stat3KO mice develop hyperplasia after 2 weeks post-induction which are eventually cleared of their hyperplasia due to immune infiltration and immune-mediated elimination (1). Only a minority of animals eventually develop non-metastatic focal tumors after escaping immune surveillance (1). Conversely, MIC β 1KO mammary glands show minimal epithelial transformation at 2 weeks post-induction, and progress slowly to develop hyperplasia and early invasive (dormant) lesions (2). These lesions are characterised by cellular senescence, epithelial cell detachment, apoptosis, and decreased proliferation (1). β 1 integrin ablation led to a significant increase in OPN whereas Stat3 ablation led to a significant decrease in OPN. We used the MIC Stat3KO model as a functional validation that OPN is a Stat3 target.
- b) Reference 1: Jones LM, Broz ML, Ranger JJ, Ozcelik J, Ahn R, Zuo D, Ursini-Siegel J, Hallett MT, Krummel M, Muller WJ. STAT3 Establishes an Immunosuppressive Microenvironment during the Early Stages of Breast Carcinogenesis to Promote Tumor Growth and Metastasis. *Cancer Res*. 2016 Mar 15;76(6):1416-28. doi: 10.1158/0008-5472.CAN-15-2770. Epub 2015 Dec 30. PMID: 26719528; PMCID: PMC5052827.
- c) Reference 2: Bui T, Gu Y, Ancot F, Sanguin-Gendreau V, Zuo D, Muller WJ. Emergence of β 1 integrin-deficient breast tumours from dormancy involves both inactivation of p53 and generation of a permissive tumour microenvironment.

Oncogene. 2022 Jan;41(4):527-537. doi: 10.1038/s41388-021-02107-7. Epub 2021 Nov 15. PMID: 34782719; PMCID: PMC8782722.

11. Fig. S2A: The interpretation of the immunoblot for β 1 integrin is challenging, raising concerns about the efficacy of β 1 integrin knockout. Evaluation via flow cytometry would provide more convincing data.

- a) To address similarly as Reviewer #1, we have re-run the immunoblot for β 1 integrin to ensure clarity of the blot. The β 1 integrin band is around 130 kDa.

12. Fig. S2E,F: Given the study's focus, it is crucial to investigate the effects of OPN on the proliferation of MIC β 1KO tumor cells.

- a) We agree that validating OPN's effect on MIC β 1KO epithelial cells' proliferation would further support our study. Unfortunately, after numerous attempts at dissociating MIC β 1KO tumors and subsequently in vitro culture, these cells repeatedly fail to establish and die after 2-4 passages. Similarly, MMTV-PyV mT *Itgb1* fl/fl cell lines on which we performed AdCre-mediated β 1 integrin excision cannot be passaged and re-plated after β 1 integrin excision for proliferation assay either. To this end, Supplementary Fig. 2e-f show that adding OPN to MMTV-PyV mT cells increases their proliferation and Fig. 2g-h shows that exogenous OPN increases cell proliferation (Ki67+) in MIC β 1KO mice.

13. Fig. 2G: Enhancing the visibility of Beige dots by changing their color and providing higher magnification images would aid readers in data assessment. Clarifying the reference point for the 'percentage of PanCK+ Ki67+ cells' would enhance data understanding. Additionally, comparing the percentage of Ki67+PanCK+ cells within total PanCK+ cells located <50 μ m from macrophages to that within total PanCK+ cells located >50 μ m away would offer a comprehensive perspective.

- a) We have updated Fig. 3g (previously Fig. 2g) so that the colours are individually presented first, which should help with the visualization. We have clarified in the revised manuscript that the "percentage of PanCK+ Ki67+ cells" refers to the total number of cells, firstly gated for PanCK positivity and then for Ki67 expression. We have also added the percentage of non-proliferating PanCK+ Ki67- cells located <50 μ m from the nearest F4/80+ CD206+ macrophages (Fig. 3i). Since the remaining percentage of both groups are located >50 μ m from macrophages, we decided to not include the respective graphs into Fig. 3 (previously Fig. 2).

14. Fig. 4, Fig. 5: Please clarify the reference point for the percentage values.

- a) Unless otherwise specified, we have chosen to consistently use % Marker+ cells reflective of the percentage of total analyzed cells. For example, in Fig 2f and Fig. 2h, "% Ki67+ cells" refers to the percentage of total cells analyzed that express Ki67. We have clarified that in the Materials and Methods section of the revised manuscript.

15. Fig. S8A,B: The description of the data is difficult to comprehend. Clarification would improve understanding.

a) We have modified the figure description in the revised manuscript.

16. Fig. 8, Fig S10: While I understand the challenges in accessing clinical samples, comparing primary and metastatic tumors may not be suitable for this study. Instead, focusing on local recurrence tumors would provide a more relevant comparison with the mouse data.

a) We fully agree with Reviewer #2 that thorough analyses on patient-matched primary breast tumors and local recurrent breast tumors would be an invaluable addition to our study. It would be the most appropriate clinical samples where we can validate our *in vivo* results. Unfortunately, obtaining such sets of samples is of significant challenge. Patient-matched primary and local recurrent tumors proved to be exceptionally difficult to obtain due to their limited availability and logistical constraints. Additionally, Therefore, we collaborated with Dr. Rinath Jeselsohn from the Dana Faber Cancer Institute who generously provided the samples we carefully analyzed in our manuscript.

REVIEWER #3

- 1. At the beginning, it is not clear why the integrin B1 KO was used / introduced. This becomes more confusing about halfway through the results with the sentence “It is well established that OPN can stimulate cell migration, mainly by engaging with integrin receptors [7].” Is the idea that integrin B3 is regulating OPN, or that when integrin B3 is knocked out OPN cannot signal through it anymore? Since many of the experiments presented are on the MIC B1 KO background, the rationale for its use at the beginning needs to be better explained.**
 - a. We chose to use the MIC β 1KO transgenic mouse model to recapitulate breast cancer recurrence stems from our previous study characterizing this model where we observed that ablation of β 1 integrin in the mammary epithelium resulted in a senescence-driven phenotype (1). MIC β 1KO tumors eventually escaped said dormancy to recur (1). As addressed for Reviewer #2’s Q9, one of the features of MIC β 1KO recurrent tumors is their distinct remodelling of acellular stromal tumor environment, with a notable increase in ECM deposition. We further pinpointed OPN as one of the ECMs contributing to TME-driven recurrence, leading to this manuscript. Furthermore, we provided a parallel explanation in the manuscript introduction: “Previously, we demonstrated that mammary epithelium-targeted disruption of β 1 integrin in a doxycycline-inducible Polyomavirus middle T antigen-driven mouse model of breast cancer (“MIC” mouse) dramatically impaired mammary tumor development via senescence-mediated dormancy [3, 5, 6]. Yet, after a variable period of dormancy (6-28 weeks), 70% of MIC β 1KO mice developed recurrent mammary tumors [3]. Therefore, this model closely recapitulates many salient features of human breast cancer dormancy and recurrence.”
 - b) Reference (1): Bui T, Gu Y, Ancot F, Sanguin-Gendreau V, Zuo D, Muller WJ. Emergence of β 1 integrin-deficient breast tumours from dormancy involves both inactivation of p53 and generation of a permissive tumour microenvironment. *Oncogene*. 2022 Jan;41(4):527-537. doi: 10.1038/s41388-021-02107-7. Epub 2021 Nov 15. PMID: 34782719; PMCID: PMC8782722.
- 2. The data suggesting OPN increases tumor proliferation through an autocrine or paracrine effect is not convincing. Supp2E does not match with the statistics – it is showing a highly significant effect of rmOPN on proliferation but appears to not change confluency based on the y-axis scale. In 2F OPN also does not appear to have much effect on proliferation. Later data in Figure 4 are more convincing that OPN increases proliferation with the Ki67 staining, but the effect is not consistent throughout the manuscript.**
 - a. We apologize that Supplementary Fig. 2e-f may have been confusing and have now modified to only keep the relevant concentration of recombinant mouse osteopontin

(25 µg/mL) for the proliferation assay on MMTV-PyV mT cells. Overall, both our *in vitro* (Supplementary Fig. 2e-f) and *in vivo* data (Fig. 2) remain concurrent that OPN does stimulate PyV mT-driven epithelial cell proliferation.

3. **The main hypothesis seems to be that OPN is secreted by the tumor cells, which increases recruitment of macrophages with OPN receptors, polarizes them to be pro-tumorigenic, which then supports tumor growth and exit from dormancy. However, there are no measures of dormancy or proliferation (Ki67, mitotic index, p27, p16, etc) in the tumor cells in the early studies (again these come up in Figure 4, but the earlier data is weaker by not including these markers). Would it not be more direct to knockdown the elevated OPN observed in the B1 KO and show that this reverses the increase in OPN receptor+ macrophages?**
 - a. We have previously completed a comprehensive set of analyses on dormancy comparing the MIC β 1KO mice to MIC WT counterparts and found that it is mainly driven by cellular senescence. This includes β -galactosidase activity, cleaved caspase 3, Ki67, p16INK4a, p53, cyclin D1, and Rb (1). Deletion of OPN in MIC WT and in MIC β 1KO mice would indeed complement this study, followed by a full paneled histological and immune analyses. We have already ordered the conditional B6.Spp1fl-EGFP-stop-tdTomato OPN KO mouse generated by Dr. Tatjana Jakobs, which we plan on back-crossing to the FVB, then introducing the MIC and β 1fl/fl alleles (2-3). These new crosses will likely take an additional 1.5-2 years to complete and will become the independent follow-up manuscript to this work.
 - b. Reference 1: Bui T, Gu Y, Ancot F, Sanguin-Gendreau V, Zuo D, Muller WJ. Emergence of β 1 integrin-deficient breast tumours from dormancy involves both inactivation of p53 and generation of a permissive tumour microenvironment. *Oncogene*. 2022 Jan;41(4):527-537. doi: 10.1038/s41388-021-02107-7. Epub 2021 Nov 15. PMID: 34782719; PMCID: PMC8782722 (Fig. 2). We fully agree that epithelial-specific
 - c. Reference 2: JAX stock #038709
 - d. Reference 3: Li S, Jakobs TC. Secreted phosphoprotein 1 slows neurodegeneration and rescues visual function in mouse models of aging and glaucoma. *Cell Rep*. 2022 Dec 27;41(13):111880. doi: 10.1016/j.celrep.2022.111880. PMID: 36577373; PMCID: PMC9847489.
4. **The effect of a-OPN on the tumors seems to drive the reduction in tumor growth with a-OPN + a-PD-1. This does not necessarily support the idea that a-OPN re-sensitizes the tumors to a-PD-1 since they do not appear to grow slower than a-OPN treatment alone. The authors suggest that targeting OPN may be a “viable therapeutic strategy to improve immunotherapy response, but they have not demonstrated this. This conclusion would need to be supported by data demonstrating that there is increased T-cell-mediated killing of the tumor cells in vivo when a-OPN and a-PD-1 are**

combined, and the effect would need to be greater than that of a-PD-1 alone and a-OPN alone. An increase in T cell infiltration is not sufficient to indicate that T cell-mediated killing is happening (e.g., they may be exhausted).

- a. We understand Reviewer #3's concern with regards to the anti-PD-1/anti-OPN combinational therapy. The reason why we stated that anti-OPN re-sensitizes tumors to anti-PD-1 treatment is because these tumors are resistant to anti-PD-1 monotherapy as shown previously (1-4). Anti-OPN treatment did have an effect alone but the effect amplified in the presence of anti-PD-1. This difference is statistically significant when comparing the combinational treatment to anti-PD-1 monotherapy which is why we used the phrase "re-sensitize". We mismatched the p values in the original panel, and the correct significance is $p = 0.0156$ between IgG control and anti-PD-1 and anti-mOPN combined, $p = 0.0457$ between IgG control and anti-mOPN alone, and no significance between IgG control and anti-PD-1 alone. This supports Reviewer #3's statement that the effect of the combinational treatment is greater than that of anti-PD-1 or anti-mOPN alone. We sincerely apologize and have corrected this in the revised figures. In order to better demonstrate the role of T cell infiltration, we performed fluorescent IHC staining for cytotoxic T cell activation markers which demonstrated that the combination treatment led to a significant improvement in T cell recruitment and activation. This increase was evident for anti-OPN monotherapy but was stronger in the group that received the combination of anti-OPN and anti-PD-1 (Supplementary Fig. 9) which mimics the tumor kinetic curve. In addition, anti-PD-1 monotherapy had no effect on T cell number of activation, further reinforcing the limited impact it has on this model. Our previous work using the PyV mT as well as the MIC mouse models has demonstrated that such increase in T cell number and activation was directly linked to tumor cell death and slower tumor growth. We have cited these articles in our paper to reinforce the functionality of T cells in this model (4-5). We have adjusted the revised manuscript to state that the combinational treatment led to an improved response and that the tumors exhibited slower growth. We do understand that the effect of this combinational treatment was more additive rather than synergistic given OPN's effectiveness as a monotherapy. We have therefore corrected this in the revised manuscript to better reflect the data.
- b. Reference 1: Masenheimer D.J. Jensen S.M. Afentoulis M.E. Wegmann K.W. Feng Z. Friedman D.J. Gough M.J. Urba W.J. Fox B.A. Timing of PD-1 blockade is critical to effective combination immunotherapy with anti-OX40. Clin. Cancer Res. 2017; 23: 6165-6177
- c. Reference 2: Li Q. Wang Y. Jia W. Deng H. Li G. Deng W. Chen J. Kim B.Y.S. Jiang W. Liu Q. et al. Low-dose anti-angiogenic therapy sensitizes breast cancer to PD-1 blockade. Clin. Cancer Res. 2020; 26: 1712-1724

- d. Reference 3: Shen M. Smith H.A. Wei Y. Jiang Y.Z. Zhao S. Wang N. Rowicki M. Tang Y. Hang X. Wu S. et al. Pharmacological disruption of the MTDH–SND1 complex enhances tumor antigen presentation and synergizes with anti-PD-1 therapy in metastatic breast cancer. *Nat. Cancer*. 2022; 3: 60-74
 - e. Reference 4: Taifour T. Attalla S.S. Zuo D. Gu Y. Sanguin-Gendreau V. Proud H. Solymoss E. Tung B. Kuasne H. Papavasiliou V. et al. The tumor-derived cytokine Chi311 induces neutrophil extracellular traps that promote T cell exclusion in triple-negative breast cancer. *Immunity*. 2023; 56: 2755-2772
 - f. Reference 5: Jones L.M. Broz M.L. Ranger J.J. Ozcelik J. Ahn R. Zuo D. Ursini-Siegel J. Hallett M.T. Krummel M. Muller W.J. STAT3 Establishes an Immunosuppressive Microenvironment during the Early Stages of Breast Carcinogenesis to Promote Tumor Growth and Metastasis. *Cancer Res*. 2016; 76: 1416-1428
5. **The finding that T cell infiltration increases with OPN treatment is interesting and striking in the images. Does PD-L1 expression in the tumor change? Understanding the status of PD-L1 is important for confirming the mechanism. Are the T cells functional, or do they express exhaustion markers (e.g. Tim3, Lag3)?**
- a. We performed fluorescent IHC staining for PD-L1 on tumors and mammary glands of MIC WT mice treated with IgG or anti-mOPN, and the percentages of total PD-L1+ cells and PD-L1+ PanCK+ cells are not significantly higher in anti-mOPN-treated mice (Supplementary Fig. 8g-i). We also performed fluorescent IHC staining for T cell exhaustion marker Tim3, which did not show any differences between treated and control groups (Supplementary Fig. 8j-l). Additionally, we added T cell activation markers granzyme B by fluorescent IHC and interferon-gamma by RNA Scope (*Ifng*), both of which are significantly higher in anti-mOPN-treated mice (Supplementary Fig. 8a-f). Overall, these data demonstrate that the T cells are functional and not exhausted after anti-mOPN treatment. The revised manuscript has been modified to incorporate these changes accordingly.
6. **The in vivo data does not directly demonstrate the role of OPN in tumor recurrence, so need to tone down statements like “in perfect concordance with our in vivo data” in reference to the patient data in Figure 8. In Figure 8B, it is difficult to determine how much PanCK+ tumor overlaps with OPN. Please add a panel of PanCK+ cells only (not merged with OPN) and quantitation of the % OPN+ PanCK+ cells as have been done for other figures. While Fig 8D&E are supportive, the correlation of CD68+ cells with OPN+ cells is quite weak (as is 9C), and does not fit with the lack of a change in CD68+ macrophages in the metastatic vs primary tumor samples (Fig 8C). The patient data strongly support a potential role for OPN in metastatic recurrence, but not through an increase in macrophage infiltration.**
- a. We have modified the text to more accurately reflect our data as per Reviewer #3’s suggestions. We also included an individual PanCK panel for the fluorescent IHC,

as well as the quantification for OPN+ PanCK+ cells (Figure 8b-e). Furthermore, we fully agree that the number of CD68+ macrophages in recurrent metastasis tumors did not significantly increase compared to their primary breast tumor counterparts. However, we do want to highlight that there is a significant positive correlation between CD68+ macrophage infiltration and OPN+ cells in these recurrent metastatic tumors (Supplementary Fig. 10e). We kept this clarification in the Results section of the revised manuscript.

- 7. The authors proposed working model does not fit with the proposed model from Weinberg in colleagues in 2008, where they proposed that OPN is secreted by the primary tumor and that promotes recurrence in the secondary site by mobilizing cells in the bone marrow. The patient data presented herein demonstrate that OPN is expressed at higher levels in the metastatic site. Please include more on this in the discussion and place the findings in the context of the field (they do not have to agree but need to be discussed more in depth to provide context).**
 - a. We have included in our revised manuscript that the increase in OPN-expressing epithelial cancer cells in metastatic tumors does not rule out the possibility that OPN is systemically elevated in these patients. Indeed, several studies, including McAllister SS *et al.* Cell. 2008, demonstrated that OPN promotes metastatic recurrence by establishing a permissive metastatic niche through other stromal cells like myeloid-derived suppressor cells and bone marrow cells prior to clinically detectable metastases. We have added these citations in our revised manuscript.
- 8. The conclusions are strongly stated and sometimes overstated throughout the results and discussion. For example, “Collectively, these data argue that increased OPN secretion by mammary epithelial tumor cells within recurrent tumors provides the necessary tumor cell autonomous and microenvironmental support for cancer recurrence. This includes restructuring an immune and fibrotic TME landscape through OPN-responsive pro-tumorigenic macrophages” – the authors have not conclusively demonstrated this. There are several strong statements with regard to the patient data validating the in vivo data, but these do not align perfectly and should be toned down. For example, “Unquestionably, our extensive patient data underscore the relevance and accuracy of our findings in vivo.”**
 - a. We have modified the statements in the revised manuscript accordingly.
- 9. Please clarify in the methods whether the MIC Stat3 KO mouse is a global KO or a mammary epithelium specific Stat3 KO. Based on the text I believe it was mammary epithelium-specific, but could be clarified.**
 - a. We clarified in the revised manuscript that it is a mammary epithelium-specific deletion of Stat3 (1).
 - b. Reference 1: Jones LM, Broz ML, Ranger JJ, Ozcelik J, Ahn R, Zuo D, Ursini-Siegel J, Hallett MT, Krummel M, Muller WJ. STAT3 Establishes an Immunosuppressive Microenvironment during the Early Stages of Breast

Carcinogenesis to Promote Tumor Growth and Metastasis. Cancer Res. 2016 Mar 15;76(6):1416-28. doi: 10.1158/0008-5472.CAN-15-2770. Epub 2015 Dec 30. PMID: 26719528; PMCID: PMC5052827.

10. Western blots in Supp 2A are not convincing for increased pStat3 or Stat3, and do not appear to match with panel B.

- a. We addressed a similarly valid concern by Reviewer #1 Q2, where we do agree that the total Stat3 and the p-Stat3 bands are not as obviously strong in the AdCre MMTV-PyV mT *Itgb1* fl/fl cell lines by eye, however, we did normalize each band to their respective control (β -actin), the ratio of which is quantified.

11. The IL-4/PanCK co-localization is difficult to see. Can the channels be two more distinct colors? They are too similar in color to appreciate co-localization.

- a. We changed the color of *IL-4*'s RNA scope signal in Fig. 3d to yellow for higher contrast with PanCK (purple) to help with visualization.

12. Figure 2 G-H could also indicate that the tumor cells secrete more chemoattractants for the F4/80+ macrophages. Is MCP-1 elevated?

- a. We performed a cytokine array (1) on MIC WT and recurrent MIC β 1KO tumors, showing that MCP-1 is comparable between the two groups which we excluded from the revised figures for this study.
- b. Reference: Eve Technologies, MOUSE CYTOKINE/CHEMOKINE 44-PLEX DISCOVERY ASSAY® ARRAY (MD44)

13. I do not think autocrinally is a word.

- a. We apologize for this and have removed it in the revised manuscript.

14. CD44 does co-localize with F4/80+ cells, but presumably also with PanCK+ tumor cells, which are missing from Supp 4E.

- a. It is correct that a significant proportion of tumor epithelial cells in recurrent MIC β 1KO tumors also express CD44. In order to quantify CD44 expression in the epithelial compartment of the tumors, we used the Classifier function in HALO 2.0 software (Indica Lab), the percentage of CD44+ cells in the epithelial-tumor compartment is significantly higher ($p = 0.0132$) in the recurrent MIC β 1KO tumors. To simultaneously address Reviewer #2-Q7, we re-stained the tissues with CD206 and quantified the percentage of OPN receptor-expressing pro-tumorigenic macrophages (% F4/80+ CD206+ with either CD44+ or β 3 integrin+) in Supplementary Fig. 5e-j.

15. Supp 5B is too small to read.

- a. As per Reviewer #4 Q1's suggestions, we have now removed Fig. 5b and adjusted the text accordingly in the revised manuscript.

16. The quantification for Supp 7B does not match the image in panel A.

- a. We apologize for the mismatch and have included a more representative image of the fluorescent IHC for Supplementary Fig. 7c (previously Supplementary Fig. 7a).

17. Please clarify the findings of Fig 9B in the text – the figure panel is confusingly presented.

- a. We simplified the finding interpretation for Fig. 9b in the text with more details in the corresponding figure legend in the revised manuscript (1).
- b. Reference 1: Abravanel DL, Belka GK, Pan TC, Pant DK, Collins MA, Sterner CJ, Chodosh LA. Notch promotes recurrence of dormant tumor cells following HER2/neu-targeted therapy. *J Clin Invest.* 2015 Jun;125(6):2484-96. doi: 10.1172/JCI74883. Epub 2015 May 11. PMID: 25961456; PMCID: PMC4497740.

18. Define EMC in the discussion.

- a. We have modified the statements in the revised manuscript accordingly and added a definition in the introduction.

19. In the methods mouse tissue collection describes bone samples that were collected and decalcified for histomorphometry but these data are not included. Are there bone metastasis samples that are available and could be reported?

- a. We apologize for the confusion. We did collect bone samples of mice who received exogenous recombinant mouse OPN or control and of mice who received anti-OPN neutralizing antibody or control for histopathological analyses. No differences were seen and these data were not included in this manuscript. We have now removed this from the Materials and Methods section of the revised manuscript.

20. Please include more detail under human samples in the methods on how they were stained.

- a. We clarified in the Materials and Methods section of the revised manuscript that human samples were stained following the same protocol as for mouse samples. Human-specific antibodies are also listed.

REVIEWER #4

2. **“we used CellPhoneDB to visualize all cell clusters that express OPN receptor complexes in response to epithelial cell (Cluster 0)-secreted OPN [17]. ...” Recent studies have shown that CellPhoneDB and CellChat have very low specificity in detection of cell 2 cell communications (<https://www.biorxiv.org/content/10.1101/2023.09.18.558298v4>). I would suggest to remove these results.**
 - a. We thank Reviewer #4 for this insightful comment and have now removed this panel and adjusted the text accordingly in the revised manuscript.
3. **“We further analyzed Kaplan-Meier survival curves generated from dataset GSE58644, ...” Please also report the HR+95% confidence interval in addition to p value.**
 - a. We have added the 95% CI to the graph on the revised Figures.

Once again, we would like to sincerely thank our Reviewers for the insightful comments and feedback. We are confident that this strengthened, revised manuscript successfully addresses your concerns to ensure that we provide a satisfactory research article for the readership of *Nature Communications*.

REVIEWERS' COMMENTS

Reviewer #1 (Remarks to the Author):

The authors addressed this reviewer's comments by clarifications and reference citations. Some of the comments were addressed by reanalysis of the data. Only one comment was not fully addressed due to that they "currently do not possess any fresh recurrent β 1KO tumors". Overall, I have no further comments for this revised manuscript.

Reviewer #2 (Remarks to the Author):

The authors have resubmitted a revised and improved manuscript to address the major concerns raised by reviewer.

My primary concern was the lack of evidence indicating that the increase in OPN is necessary for the recurrence of mammary tumors in MIC β 1KO mice. The authors addressed this by presenting data showing that treatment with an anti-OPN antibody suppressed the growth of recurred tumors isolated from MIC β 1KO mice and orthotopically transplanted into recipient mice (Fig. S7). Additionally, they demonstrated that pharmacological depletion of macrophages also suppresses the growth of these transplanted tumors (Fig. 5). These findings suggest that increased OPN levels and macrophage recruitment promote the outgrowth of "recurred" tumors. However, it is already well-established in the literature that OPN and macrophages promote tumor growth. While the authors acknowledged the challenges of the MIC β 1KO model, further investigation into the effects of OPN and recruited macrophages on the dormancy period would enhance the manuscript's value.

The data presented in the revised manuscript (Fig. 4, Fig. S3, Fig. S4) are not convincing enough to support the statement that OPN, together with IL-4, regulates macrophage polarization. The data in revised Fig. S5 still do not rule out the possibility of an increase in a TAM subpopulation stably expressing OPN receptors. While the data indicate that F4/80+CD206+ cells co-express CD44 and integrins, this does not necessarily imply an increase in CD44/integrin expression within individual macrophages.

Reviewer #3 (Remarks to the Author):

The authors have done a thorough job of responding to critiques and made substantial changes in response to reviewers. I have a few remaining comments/concerns:

1. The in vitro effect of rOPN on tumor proliferation remains unconvincing. While the changes are statistically significant, they do not appear biologically significant (Supp fig 2e-f). The effect in vivo in WT ice is far more convincing and suggests that there are other microenvironmental factors involved.
2. The new in vivo crosses sound interesting, but could you not just knockout OPN in MIC B1KO cancer cells cultured ex vivo then inject the cancer cells in vivo and see if that reverses the increase in OPNR+ macrophages? This would also directly get at the relative importance of tumor-derived vs host-derived OPN .
3. The authors state that "Anti-OPN treatment did have an effect alone but the effect amplified in the presence of anti-PD-1. This difference is statistically significant when comparing the combinational treatment to anti-PD-1 monotherapy which is why we used the phrase "re-sensitize". The issue is not the comparison between the combination therapy and anti-PD-1 monotherapy. The issue is that there is no significant difference between anti-OPN and anti-OPN + anti-PD-1. Why treat with anti-PD-1 at all if anti-OPN is sufficient to limit tumor growth? Anti-PD-1 has high toxicity, so the justification for using anti-OPN to re-sensitize tumors to anti-PD-1 does not make sense to me, when you could just use anti-OPN alone and achieve the same effect.

Dear Reviewers,

We thank all the Reviewers for your continued interest and insightful feedback on our manuscript entitled '*Tumor-derived osteopontin is a therapeutic target that drives breast cancer recurrence.*' Here, we would like to thoroughly respond on each of your comments.

REVIEWERS' COMMENTS

Reviewer #1 (Remarks to the Author)

The authors addressed this reviewer's comments by clarifications and reference citations. Some of the comments were addressed by reanalysis of the data. Only one comment was not fully addressed due to that they "currently do not possess any fresh recurrent β 1KO tumors". Overall, I have no further comments for this revised manuscript.

Reviewer #2 (Remarks to the Author)

The authors have resubmitted a revised and improved manuscript to address the major concerns raised by reviewer.

My primary concern was the lack of evidence indicating that the increase in OPN is necessary for the recurrence of mammary tumors in MIC β 1KO mice. The authors addressed this by presenting data showing that treatment with an anti-OPN antibody suppressed the growth of recurred tumors isolated from MIC β 1KO mice and orthotopically transplanted into recipient mice (Fig. S7). Additionally, they demonstrated that pharmacological depletion of macrophages also suppresses the growth of these transplanted tumors (Fig. 5). These findings suggest that increased OPN levels and macrophage recruitment promote the outgrowth of "recurred" tumors. However, it is already well-established in the literature that OPN and macrophages promote tumor growth. While the authors acknowledged the challenges of the MIC β 1KO model, further investigation into the effects of OPN and recruited macrophages on the dormancy period would enhance the manuscript's value.

The data presented in the revised manuscript (Fig. 4, Fig. S3, Fig. S4) are not convincing enough to support the statement that OPN, together with IL-4, regulates macrophage polarization. The data in revised Fig. S5 still do not rule out the possibility of an increase in a TAM subpopulation stably expressing OPN receptors. While the data indicate that F4/80+CD206+ cells co-express CD44 and integrins, this does not necessarily imply an increase in CD44/integrin expression within individual macrophages.

Reviewer #3 (Remarks to the Author)

The authors have done a thorough job of responding to critiques and made substantial changes in response to reviewers. I have a few remaining comments/concerns:

1. The *in vitro* effect of rOPN on tumor proliferation remains unconvincing. While the changes are statistically significant, they do not appear biologically significant (Supp fig 2e-f). The effect *in vivo* in WT mice is far more convincing and suggests that there are other microenvironmental factors involved.
2. The new *in vivo* crosses sound interesting, but could you not just knockout OPN in MIC B1KO cancer cells cultured *ex vivo* then inject the cancer cells *in vivo* and see if that reverses the increase in OPN⁺ macrophages? This would also directly get at the relative importance of tumor-derived vs host-derived OPN .
3. The authors state that “Anti-OPN treatment did have an effect alone but the effect amplified in the presence of anti-PD-1. This difference is statistically significant when comparing the combinational treatment to anti-PD-1 monotherapy which is why we used the phrase "re-sensitize". The issue is not the comparison between the combination therapy and anti-PD-1 monotherapy. The issue is that there is no significant difference between anti-OPN and anti-OPN + anti-PD-1. Why treat with anti-PD-1 at all if anti-OPN is sufficient to limit tumor growth? Anti-PD-1 has high toxicity, so the justification for using anti-OPN to re-sensitize tumors to anti-PD-1 does not make sense to me, when you could just use anti-OPN alone and achieve the same effect.

REVIEWERS' COMMENTS: Point-by-point response

REVIEWER #1

1. The authors addressed this reviewer's comments by clarifications and reference citations. Some of the comments were addressed by reanalysis of the data. Only one comment was not fully addressed due to that they "currently do not possess any fresh recurrent β 1KO tumors". Overall, I have no further comments for this revised manuscript.
 - a. We thank Reviewer #1 for their insightful comments and help to ameliorate our manuscript.

REVIEWER #2

1. The authors have resubmitted a revised and improved manuscript to address the major concerns raised by reviewer. My primary concern was the lack of evidence indicating that the increase in OPN is necessary for the recurrence of mammary tumors in MIC β 1KO mice. The authors addressed this by presenting data showing that treatment with an anti-OPN antibody suppressed the growth of recurred tumors isolated from MIC β 1KO mice and orthotopically transplanted into recipient mice (Fig. S7). Additionally, they demonstrated that pharmacological depletion of macrophages also suppresses the growth of these transplanted tumors (Fig. 5). These findings suggest that increased OPN levels and macrophage recruitment promote the outgrowth of "recurred" tumors. However, it is already well-established in the literature that OPN and macrophages promote tumor growth. While the authors acknowledged the challenges of the MIC β 1KO model, further investigation into the effects of OPN and recruited macrophages on the dormancy period would enhance the manuscript's value.
 - a. We thank Reviewer #2 for their detailed feedback. To address this interest previously raised by Reviewer #2, we did further interrogate the OPN kinetics during different stages of MIC β 1KO tumor progression. Specifically, our revised manuscript includes immunostaining data showing that MIC β 1KO dormant tumors do not significantly differ in OPN protein levels compared to MIC WT counterparts (Supplementary Fig. 2) and our *in vivo* results demonstrate that OPN alone is insufficient to support MIC β 1KO tumors bypassing their primary dormancy phase (Fig. 2). Additionally, the levels of macrophages (total and CD206+) did not differ between MIC β 1KO and MIC WT mice at the dormant (early invasive) stage of

tumor progression (Fig. 5). These data indicate that OPN and macrophages promote tumor growth most significantly during recurrence and that tumor dormancy (maintenance and escape) is regulated by other mechanisms (previously published with the MIC β 1KO model (1)). Therefore, we focused the aim of this manuscript on the recurrent stage of MIC β 1KO tumorigenesis. Our results indicate that OPN is a therapeutic target in recurrent tumors, especially given its elevated levels in recurrent patient samples (Fig. 8). We do agree with Reviewer #2 that in-depth analyses of the tumor environmental adaptations including OPN and macrophages during tumor dormancy would be of great value in alternative *in vivo* models of breast tumor dormancy (2, 3) given the challenges of working with the MIC β 1KO model, although this would fall outside of the scope of this manuscript.

- b. Reference 1: Bui T, Gu Y, Ancot F, Sanguin-Gendreau V, Zuo D, Muller WJ. Emergence of β 1 integrin-deficient breast tumours from dormancy involves both inactivation of p53 and generation of a permissive tumour microenvironment. *Oncogene*. 2022 Jan;41(4):527-537. doi: 10.1038/s41388-021-02107-7. Epub 2021 Nov 15. PMID: 34782719; PMCID: PMC8782722.
 - c. Reference 2: Gu Y, Bui T, Muller WJ. Exploiting Mouse Models to Recapitulate Clinical Tumor Dormancy and Recurrence in Breast Cancer. *Endocrinology*. 2022 Jun 1;163(6):bqac055. doi: 10.1210/endo/bqac055. PMID: 35560214.
 - d. Reference 3: Bushnell GG, Deshmukh AP, den Hollander P, Luo M, Soundararajan R, Jia D, Levine H, Mani SA, Wicha MS. Breast cancer dormancy: need for clinically relevant models to address current gaps in knowledge. *NPJ Breast Cancer*. 2021 May 28;7(1):66. doi: 10.1038/s41523-021-00269-x. PMID: 34050189; PMCID: PMC8163741.
2. The data presented in the revised manuscript (Fig. 4, Fig. S3, Fig. S4) are not convincing enough to support the statement that OPN, together with IL-4, regulates macrophage polarization. The data in revised Fig. S5 still do not rule out the possibility of an increase in a TAM subpopulation stably expressing OPN receptors. While the data indicate that F4/80+CD206+ cells co-express CD44 and integrins, this does not necessarily imply an increase in CD44/integrin expression within individual macrophages.
 - a. We understand Reviewer #2's concerns on the macrophage polarization potential of OPN and IL-4. In our *in vivo* tissue characterization, we acknowledge that other factors, epithelial- and stromal-derived, may influence the polarization status of macrophages in addition to OPN which could contribute to the lack of significance in F4/80+ CD206+ macrophages in recombinant OPN-treated mice. We understand that the *in vitro* results only demonstrate that OPN and IL-4 further increase the expression of IL-4R on BMDMs than either OPN or IL-4 alone and cannot fully support this to be enough for macrophages to become pro-tumorigenic. Rather, OPN and IL-4 can further accentuate some contributing factors of a pro-tumorigenic state in macrophages, such as increasing IL-4R expression. We would

be more than happy to modify the manuscript to better reflect the interpretation of these data.

- b. Furthermore, we agree with Reviewer #2 that our data in Supplementary Fig. 5 does not explicitly show an increase in OPN receptors on individual macrophages. Although tracking the expression of OPN receptors on macrophages within tumors as it evolves through dormancy and recurrence would be highly informative, the exact molecular pathways activated in macrophages by OPN, including its own receptors, are outside the scope of this study as we similarly addressed for Reviewer #1 previously. Here, the expression of OPN receptors on macrophages highlights how OPN promotes their recruitment complemented by the migration assay (Fig. 4) and that these OPN receptor+ macrophages collaboratively support tumor growth as evidenced by the results after their depletion (Fig. 5).

REVIEWER #3

1. The authors have done a thorough job of responding to critiques and made substantial changes in response to reviewers. I have a few remaining comments/concerns: The *in vitro* effect of rOPN on tumor proliferation remains unconvincing. While the changes are statistically significant, they do not appear biologically significant (Supp fig 2e-f). The effect *in vivo* in WT ice is far more convincing and suggests that there are other microenvironmental factors involved.
 - a. We appreciate Reviewer #3's comments and feedback to improve our manuscript. We understand the concern that the *in vitro* proliferation assay only provides marginal support compared to the effects seen *in vivo*, and we would be happy to only keep the *in vivo* data in a revised manuscript.
2. The new *in vivo* crosses sound interesting, but could you not just knockout OPN in MIC B1KO cancer cells cultured *ex vivo* then inject the cancer cells *in vivo* and see if that reverses the increase in OPNR+ macrophages? This would also directly get at the relative importance of tumor-derived vs host-derived OPN .
 - a. We appreciate Reviewer #3's valuable experimental suggestions and fully agree that this is a clever approach to investigate the relative contribution of tumor epithelial- vs host-derived OPN in tumorigenesis and macrophage function. Unfortunately, *in vitro* culture of dissociated MIC β 1KO tumors cannot be established as these cells die after 2-4 passages (while on doxycycline). Similarly, MMTV-PyV mT *Itgb1* fl/fl cell lines post-AdCre-mediated β 1 integrin excision cannot be passaged and re-plated either after β 1 integrin excision. Therefore, it is experimentally unfeasible to knockout OPN in MIC β 1KO cancer cells *in vitro*. Additionally, MIC cells that are established on plastic do not grow *in vivo* after

mammary fat pad injection. We believe that their *ex vivo* establishment and maintenance induce a significant cellular change such that they can no longer re-grow into tumors *in vivo*, the exact mechanisms of which are currently unknown. This also explains why all our tumor transplant experiments do not involve established cell lines cultured in plastic first. All of our previous experimental challenges led us to order the conditional OPN-deficient mouse (1, 2) that we will subsequently characterize and cross into MIC β 1KO mice for further investigation.

- b. Reference 1: JAX stock #038709
 - c. Reference 2: Li S, Jakobs TC. Secreted phosphoprotein 1 slows neurodegeneration and rescues visual function in mouse models of aging and glaucoma. *Cell Rep.* 2022 Dec 27;41(13):111880. doi: 10.1016/j.celrep.2022.111880. PMID: 36577373; PMCID: PMC9847489.
3. The authors state that “Anti-OPN treatment did have an effect alone but the effect amplified in the presence of anti-PD-1. This difference is statistically significant when comparing the combinational treatment to anti-PD-1 monotherapy which is why we used the phrase "re-sensitize". The issue is not the comparison between the combination therapy and anti-PD-1 monotherapy. The issue is that there is no significant difference between anti-OPN and anti-OPN + anti-PD-1. Why treat with anti-PD-1 at all if anti-OPN is sufficient to limit tumor growth? Anti-PD-1 has high toxicity, so the justification for using anti-OPN to re-sensitize tumors to anti-PD-1 does not make sense to me, when you could just use anti-OPN alone and achieve the same effect.
- a. We appreciate Reviewer #3’s clarification and comment and appreciate the limitation in statistical significance between anti-OPN and the combinational treatment. The rationale behind this experiment was that anti-PD-1 is now more commonly used in the clinic whereas there are currently no clinical trials that aim to target OPN therapeutically in cancer (NIH clinicaltrials.gov). While we believe this pairing of anti-OPN and anti-PD-1 may hold promise given the increase in T cell activation post-anti-OPN treatment, we acknowledge that anti-PD-1 has significant toxicities and that anti-OPN may be better used alone in our model specifically. We also acknowledge that the clinical pharmacodynamics of anti-OPN remains to be investigated outside of the scope of this study. We have decided to keep the experiment in the revised manuscript given the increase in T cell activation and as additional support of anti-OPN’s potency in suppressing tumor growth. We would be more than happy to modify the manuscript to emphasize these points, namely the lack of significant difference between anti-OPN and the combinational treatment, remove the term “re-sensitize”, focus on the tumor growth differences, and elaborate that anti-OPN may be sufficient alone without the need for anti-PD-1, especially in immune cold tumors, to avoid unnecessary toxicities for patients.

Once again, we would like to sincerely thank our Reviewers for the insightful comments and feedback. We would be more than happy to make additional edits to the manuscript to ensure that we provide a satisfactory research article for the readership of *Nature Communications*.

We thank you for your consideration and look forward to hearing from you soon.